SciPost Physics

Submission

# Solving a family of $T\bar{T}$-like theories

B. Le Floch[1*], M. Mezei[2]

**1** Philippe Meyer Institute, Physics Department, École Normale Supérieure, PSL Research University, Paris, France
**2** Simons Center for Geometry and Physics, SUNY, Stony Brook, USA
* lefloch@lpt.ens.fr

May 3, 2019

## 1 Abstract

We deform two-dimensional quantum field theories by antisymmetric combinations of their conserved currents that generalize Smirnov and Zamolodchikov's $T\bar{T}$ deformation. We obtain that energy levels on a circle obey a transport equation analogous to the Burgers equation found in the $T\bar{T}$ case. This equation relates charges at any value of the deformation parameter to charges in the presence of constant background gauge fields. We determine the initial data and solve the transport equations for antisymmetric combinations of flavor symmetry currents and the stress tensor starting from conformal field theories. Among the theories we solve are conformal field theories deformed by $J\bar{T}$ and $T\bar{T}$ simultaneously. We check our answer against results from AdS/CFT.

# 1   Introduction

Two-dimensional field theories are interesting theoretical laboratories for discovering new phenomena in quantum field theories. An exciting recent development indicates that in special situations we may control a theory flowing against the renormalization group flow: we can deform a theory by special irrelevant operators and flow towards the ultraviolet without encountering an infinite set of ambiguities that usually plague such attempts. On top of this, in some cases the resulting theory is solvable in the sense that its spectrum on $S^1 \times \mathbb{R}$ can be determined explicitly in terms of the spectrum of the undeformed theory. This has so far been achieved for the $T\bar{T}$ and $J\bar{T}$ deformed theories. In this paper, we extend these results to a large family of deformations. Below we briefly summarize what has been understood about these theories in the literature. These exciting findings provide ample motivation for this study.

It was understood in [1] that the composite operator $T\bar{T}$ is unambiguously defined in any translationally invariant field theory, because the collision limit of the point splitted operators is regular (up to derivatives). The derivation was extended in [2] to other operators, and deforming by such irrelevant operators was proposed. The spectrum of the theory was shown to obey the Burgers equation in [2,3], see also [4] for the nonrelativistic case. The spectrum of the $J\bar{T}$ deformed theory was obtained in [5], see also [6,7].

One can also arrive at the $T\bar{T}$ deformed theories from the point of view of S-matrices. This was developed in [8–10] and a realization of it as a theory of quantum gravity was proposed in [11–13]. Other work analyzing the very interesting UV behavior of the theory depending on the sign of the coupling includes [14,15].

The torus partition function of both the $T\bar{T}$ and $J\bar{T}$ obeys interesting differential equations, has nice modular properties, and is unique in the appropriate sense [12, 16–19]. Correlation functions were analyzed in [20–24].

The first holographic interpretation of $T\bar{T}$ as cutoff AdS$_3$ geometry was proposed in [14], progress in this direction is reported in [22, 25, 26]. The holographic interpretation of $J\bar{T}$

73 deformation was studied in [6, 27]. Higher dimensional generalizations in the holographic
74 context were discussed in [28–31]. These ideas were applied to the dS/dS correspondence
75 in [32]. A second holographic interpretation of a single trace version of $T\bar{T}$ with a different
76 sign for the coupling was proposed to describe $AdS_3$ embedded in a linear dilaton background
77 in [15]. Soon this approach was generalized to the single trace version of $J\bar{T}$ in [5, 33].
78 Work in this direction includes [21, 34–39]. The entanglement entropy of these holographic
79 theories was analyzed in [40, 41].

80 Deformations of supersymmetric theories were discussed in [42, 43]. $T\bar{T}$ deformed
81 theories on $S^2$ were analyzed in [30, 44]. Classical field theories deformed by $T\bar{T}$ have
82 interesting properties on their own, which were analyzed in [3, 45–48].

# 83 Organization

84 Since our arguments borrow results from a variety of sources, we perform a number of
85 checks and make several comments in the process of solving the spectrum of the deformed
86 theories. We include these at each key step in the paper. To arrive at the result fastest,
87 the reader may wish to follow the argument narrowly and skip the checks and comments,
88 and read Section 2 for the strategy of our approach, Section 3 except for Section 3.4 for
89 the solution of the theory with background gauge fields, Section 4 for the universal flow
90 equation describing a generic point in theory space, Section 6 except for Section 6.4 for the
91 solution of the spectrum, and Section 7 for conclusions and future directions.

92 The content of the rest of the paper is as follows. Section 3.4 includes our unsuccessful
93 attempt to understand deformations by higher spin (KdV) currents, Section 5 contains two
94 complementary checks of the universal equation, and Section 6.4 checks a special case of
95 the spectrum from string theory. The Appendices contain details of our conventions, the
96 worked out example of the compact free scalar, and comparison with the $J\bar{T}$ literature.

97

98 **Note added:** Results equivalent to those in Section 6.4 have been obtained independently
99 using the same methods in ongoing work [49]. We thank the authors for comparing our
100 formulas. A summary of that work appears in a coordinated submission to the arXiv [50].

# 101 2 A strategy for solving $T\bar{T}$-like theories

102 Let us take a 2d QFT on a cylinder, $S^1 \times \mathbb{R}$, which is translationally invariant in both the
103 spatial ($S^1$) and the time ($\mathbb{R}$) directions. Note that we do not require Lorentz symmetry.
104 In [1, 2] it was shown that there exist composite operators built from conserved currents in
105 any such QFT, whose expectation value factorizes in an energy eigenstate $|n\rangle$:

$$\begin{aligned}
\langle n|\epsilon^{\mu\nu} J_\mu^{(1)}(y) J_\nu^{(2)}(y)|n\rangle &\equiv \langle n|\lim_{x \to y} \epsilon^{\mu\nu} J_\mu^{(1)}(x) J_\nu^{(2)}(y)|n\rangle \\
&= \epsilon^{\mu\nu} \langle n|J_\mu^{(1)}|n\rangle \langle n|J_\nu^{(2)}|n\rangle \,,
\end{aligned} \tag{2.1}$$

106 where in the second line we deleted the arguments to emphasize that the one point functions
107 in energy eigenstates do not depend on the position of the operator. There are two
108 familiar examples of these composite operators: taking $J_z^{(1)} \equiv J$ and $J_{\bar{z}}^{(2)} \equiv \bar{J}$ in a CFT, the
109 composite operator $\epsilon^{\mu\nu} J_\mu^{(1)} J_\nu^{(2)}$ is the exactly marginal operator $J\bar{J}$, while taking $J_\mu^{(1)} \equiv T_{1\mu}$
110 and $J_\mu^{(2)} \equiv T_{2\mu}$ in any 2d QFT, the composite operator becomes what is known as $T\bar{T}$ in
111 the literature. (In our conventions, it is $-\frac{1}{2\pi^2} T\bar{T}$.)

The composite operator $\mathcal{O} \equiv \epsilon^{\mu\nu} J_\mu^{(1)} J_\nu^{(2)}$ hence defined can be used to define a one parameter family of theories,

$$\frac{d}{d\lambda} S(\lambda) = \int d^2x \ \mathcal{O}_\lambda(x) , \tag{2.2}$$

where the notation $\mathcal{O}_\lambda(x)$ serves as a reminder that the conserved currents $J_\mu^{(1,2)}$ building $\mathcal{O}$ change as $\lambda$ is changed. Using factorization, it immediately follows that the energy spectrum of this family of theories obeys

$$\frac{\partial}{\partial\lambda} E_n = L \, \epsilon^{\mu\nu} \langle n|J_\mu^{(1)}|n\rangle \langle n|J_\nu^{(2)}|n\rangle , \tag{2.3}$$

where we used the Hellmann-Feynman theorem $\frac{\partial}{\partial\lambda} E_n = \langle n|\frac{\partial}{\partial\lambda} H|n\rangle$. If we want to use this equation, we need to know the matrix elements $\langle n|J_\mu^{(1,2)}|n\rangle$. For the time component $\mu = 2$, we have $\langle n|J_2|n\rangle = Q_n/L$, where $L$ is the length of the spatial $S^1$ and $Q$ is the charge corresponding to the conserved current. If $Q$ is the charge of an internal symmetry, its value is quantized, and cannot depend on $\lambda$. This includes the case of the momentum along the spatial $S^1$, for which $Q_n = iP_n = \frac{2\pi i j_n}{L}$, where $j_n \in \mathbb{Z}$. For time-translations, $Q_n = -E_n$. One can also consider a higher spin (KdV) charge $Q$ of a 2d CFT or integrable model, in which case to get a closed set of equations we also need to write down a flow equation for $\frac{\partial}{\partial\lambda} Q_n^{(\text{higher spin})}$. We treat one such case in a separate publication.

For the spatial component $\mu = 1$ giving $\langle n|J_1|n\rangle$, we do not in general have a physical interpretation. The case of the $T\bar{T}$ deformation of a relativistic field theory is an exception, where we know the value of all matrix elements:

$$\langle n|T_{tt}|n\rangle = -\frac{E_n}{L} , \qquad \langle n|T_{xx}|n\rangle = -\partial_L E_n , \qquad \langle n|T_{xt}|n\rangle = \frac{iP_n}{L} = \langle n|T_{tx}|n\rangle , \tag{2.4}$$

where the last equality follows from the fact that the stress tensor is symmetric. Plugging these into (2.3) we obtain the Burgers equation of [2]:

$$\frac{\partial}{\partial\lambda} E_n = \frac{1}{2} \left( E_n \partial_L E_n + \frac{P_n^2}{L} \right) , \tag{2.5}$$

where the overall factor on the RHS follows from our choice of normalization of the composite operator $\mathcal{O}$, as discussed below (2.3).

We propose to proceed in the more general case, where general considerations do not determine $\langle n|J_1|n\rangle$, by coupling the current to an infinitesimal constant background field

$$\delta_a S(\lambda, a) \equiv \int d^2x \ iJ_1(x) . \tag{2.6}$$

With the introduction of $a$, (2.3) becomes:

$$\frac{\partial}{\partial\lambda} E_n = \frac{1}{L} \left( Q^{(2)} \delta_{a^{(1)}} E_n - Q^{(1)} \delta_{a^{(2)}} E_n \right) . \tag{2.7}$$

We do not want to introduce background fields for quantities that we know from other considerations, hence in such cases it is understood that $\delta_a E_n$ should be replaced by the appropriate quantity in this equation, see e.g. (2.4).

In order for (2.7) to be useful, we have to understand two things. First, to use it as an evolution equation, we need to understand the equation away from infinitesimal $a^{(I)}$. We refer to these deformations by the spatial component of a current as *turning on a background gauge field*, even when the current is part of the stress-tensor and the gauge

143  field is actually a vielbein. We work to all orders in $a^{(I)}$, not just first (linear) order. We will
144  refer to the deformations by the quadratic composite operators $\mathcal{O}$ as *bilinear deformations*;
145  again we work to all orders in $\lambda$. The deformations do not in general commute, namely
146  the vector fields describing the flow in coupling space have a nonzero Lie bracket. We
147  want to understand the flow in some coordinate system in coupling space $(\lambda, a^{(I)})$ taking
148  into account this noncommutativity. Second, if we want to solve the $\lambda$-evolution in this
149  enlarged coupling space, we need to understand the theory not just at $S(\lambda = 0)$, but at
150  $S(\lambda = 0, a^{(I)})$. This can be done if the theory at $\lambda = a^{(I)} = 0$ is a CFT, because for
151  holomorphic (antiholomorphic) currents, $J_1 = \mp i J_2$. Besides all these challenges, we have to
152  make sure that ambiguities (e.g., improvement transformations) do not ruin the universality
153  of the result. In Figure 1 we give an illustration of our strategy.

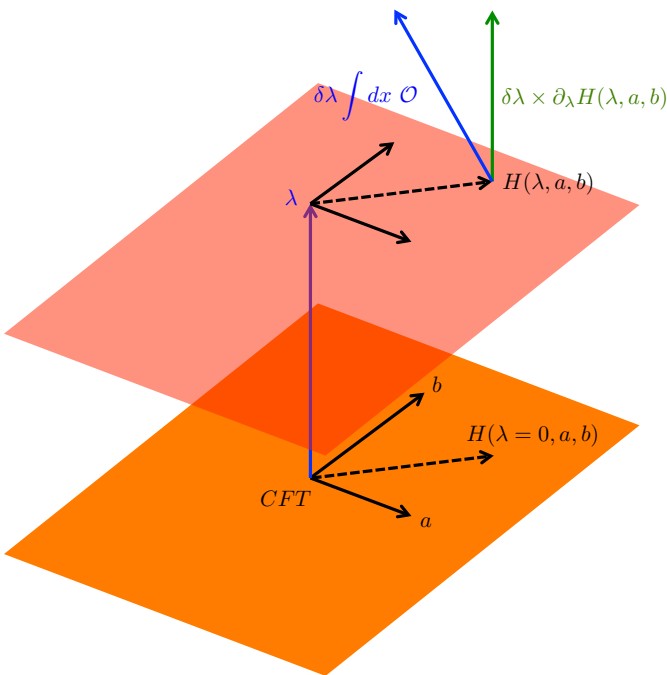

Figure 1: Graphical representation of the strategy solving deformations of CFTs by bilinear composite operators. Turning on the background gauge fields $a, b$ determines the initial value surface, drawn here as a bright orange plane. These are the directions corresponding to deformations by spatial components of currents. The $\lambda$ direction in coupling space represents the deformation by the bilinear composite operator $\mathcal{O}$. We erect a coordinate system by first deforming by $\int dx \, \mathcal{O}_\lambda$ as in (2.2) and going $\lambda$ distance. Subsequently we turn on background gauge fields. Hence, deforming a generic point in coupling space by $\delta\lambda \int dx \, \mathcal{O}$ (indicated by blue arrow) does not in general agree with $\delta\lambda \times \partial_\lambda H(\lambda, a, b)$.

154       In what follows, we present strong arguments that the outlined strategy works for a
155  large family of irrelevant deformations of CFTs. We remark that [5] solved the $J\bar{T}$-deformed
156  theory using a different method: the existence of a holomorphic current. We reproduce their
157  results in our framework. We explain how to reconcile the two viewpoints in Appendix D.

## 3    Background gauge fields

158   In what follows, we find it convenient to work in the Hamiltonian formalism on $S^1 \times \mathbb{R}$
160   with objects understood to be operators. The conservation equation of a current $J_\mu$ in our
161   conventions is:

$$0 = \partial_x J_x + [H, J_t] \,. \tag{3.1}$$

162   Our conventions are collected in Appendix A.

### 3.1    CFT deformed by stress tensor

164   Using translational invariance, we know the diagonal matrix elements of the stress tensor
165   (2.4), except for that of $T_{tx} \neq T_{xt}$ since we do not assume Lorentz invariance of the deformed
166   theory. According to the general strategy, we introduce a constant background field $b$ for
167   this operator. We want to determine $H(b)$ for finite $b$. Its evolution equation is

$$\frac{\partial}{\partial b} H(b) = -i \int dx \, T_{tx}(x) \,. \tag{3.2}$$

168   Note that in Euclidean signature $T_{tx}$ is antihermitian, hence the $i$ in the above formula. We
169   were not able to determine $H(b)$ in closed form, if we only assume Lorentz symmetry for
170   $H(b = 0)$. If $H(b = 0)$ describes a CFT, however, we obtain a solvable system of equations.
171   Of course, we only have conformal symmetry at the starting point of the flow equation
172   (3.2). Away from $b = 0$ the stress tensor will not be symmetric as can be seen from the
173   explicit expressions we give below.

174       Starting from a CFT we have $T_{tt}^{(0)} + T_{xx}^{(0)} = 0$ (at zero deformation) in addition to
175   $T_{tx}^{(0)} = T_{xt}^{(0)}$. Using the definition (3.2), we get $\partial_b T_{tt} = i T_{tx}$. Because momentum is
176   quantized and hence cannot depend on $b$, we have $\partial_b T_{xt} = 0$. Using the conservation
177   equations $\partial_x T_{\mu x} = -[H, T_{\mu t}]$, we work out $T_{xt} = T_{xt}^{(0)}$, and

$$\begin{aligned}
T_{tt} &= -T_{xx} = \frac{1}{1 - b^2} T_{tt}^{(0)} + \frac{ib}{1 - b^2} T_{xt}^{(0)}, \\
T_{tx} &= \frac{-2ib}{(1 - b^2)^2} T_{tt}^{(0)} + \frac{1 + b^2}{(1 - b^2)^2} T_{xt}^{(0)},
\end{aligned} \tag{3.3}$$

178   where we only had to use (A.4) that gives $H, P$ in terms of the components of the
179   stress tensor, and (A.5) that gives the spacetime translations they generate. Interestingly
180   $T_{tt} + T_{xx} = 0$ for all $b$. Integrating $T_{tt}, T_{xt}$ as in (A.4) we find:

$$P = P^{(0)}, \qquad H = \frac{H^{(0)} + b P^{(0)}}{1 - b^2} \,. \tag{3.4}$$

181   It will be helpful to rewrite the result for $H$ as:

$$\begin{aligned}
H &= -\left( \frac{1}{1 - b} P_1^{(0)} + \frac{1}{1 + b} P_{-1}^{(0)} \right), \\
P_{\pm 1}^{(0)} &\equiv -\frac{H^{(0)} \pm P^{(0)}}{2} \,,
\end{aligned} \tag{3.5}$$

182   which we interpret to say that the initial value of the holomorphic and antiholomorphic
183   charges contribute to $H$ weighted by the factor $\frac{1}{1 \mp b}$.

### 3.2 CFT deformed by currents and the stress tensor

We consider a CFT with left- and right-moving $U(1)$ symmetry currents $J_\mu$ and $\bar{J}_\mu$. The case of a CFT deformed by only $J_x$ is straightforward, so we move on to discussing a CFT deformed by $J_x$, $\bar{J}_x$, and $T_{tx}$. In familiar deformations of CFTs, we are used to losing one of the conserved currents. In contrast, a curious feature of the deformed theories that we consider is that the currents $J_\mu$ and $\bar{J}_\mu$ remain separately conserved.

We will see that it is possible to keep the corresponding conserved charges unchanged under the deformation. An easy way to argue for this is to take an example where the charges generate a compact $U(1) \times U(1)$ symmetry: the spectrum of charges cannot depend on the deformation due to charge quantization. Such an example is provided by the compact scalar discussed in Appendix B. Since our methods do not depend on global aspects, we then expect that the spectrum of charges corresponding to internal symmetries does not change. We will check by explicit computation that this is indeed the case.

We observe a major simplification during the derivation: turning on background gauge fields for different symmetries commutes (to all orders in the background fields $a$, $\bar{a}$, $b$). We derive this using explicit computation. An explanation for this is that all operators that feature in the derivation are neutral and hence commute with $Q$, $\bar{Q}$. Thus adding them to the Hamiltonian does not change the conservation equation, and the currents $J_\mu$, $\bar{J}_\mu$ remain unchanged under the $a$, $\bar{a}$ deformation. Noncommutativity, however, will be an essential aspect of the physics of the coupling space flow once we include bilinear deformations in Section 4.

We expect based on (3.5) that the conserved charges behave as:

$$
Q(a, \bar{a}, b) = Q^{(0)} , \qquad \bar{Q}(a, \bar{a}, b) = \bar{Q}^{(0)} , \qquad P(a, \bar{a}, b) = P^{(0)} ,
$$
$$
H(a, \bar{a}, b) = \frac{H^{(0)} + bP^{(0)}}{1 - b^2} + \frac{aQ^{(0)}}{1 - b} + \frac{\bar{a}\bar{Q}^{(0)}}{1 + b} ,
$$
(3.6)

where we note that the internal symmetry charges $Q^{(0)}, \bar{Q}^{(0)}$ are pure numbers (see Appendix B for their allowed values for the example of the compact scalar), while $P^{(0)} \in \frac{2\pi}{L}\mathbb{Z}$ depends on $L$. It is only the second line that is an Ansatz, the first line follows from general principles.

A little bit of thought leads to the following Ansatz for the currents that we will verify below:

$$
T_{tt} = \frac{1}{1 - b^2} T_{tt}^{(0)} + \frac{ib}{1 - b^2} T_{xt}^{(0)} - \frac{1}{1 - b} a J_t^{(0)} - \frac{1}{1 + b} \bar{a} \bar{J}_t^{(0)} ,
$$
$$
T_{tx} = -\frac{2ib}{(1 - b^2)^2} T_{tt}^{(0)} + \frac{1 + b^2}{(1 - b^2)^2} T_{xt}^{(0)} + \frac{i}{(1 - b)^2} a J_t^{(0)} - \frac{i}{(1 + b)^2} \bar{a} \bar{J}_t^{(0)} ,
$$
(3.7)
$$
J_x = -\frac{i}{1 - b} J_t^{(0)} , \qquad \bar{J}_x = \frac{i}{1 + b} \bar{J}_t^{(0)} .
$$

The components $J_t, \bar{J}_t, T_{xt}$ cannot depend on the background fields because of the quantization conditions (3.6), while we will not need $T_{xx}$. The Ansatz clearly obeys

$$
\partial_b T_{tt} = iT_{tx} , \qquad \partial_a T_{tt} = -iJ_x , \qquad \partial_{\bar{a}} T_{tt} = i\bar{J}_x . \tag{3.8}
$$

Let us verify that the Ansatz indeed solves the problem. The current conservation

215 equation is

$$\partial_x J_x + [H, J_t] = -\frac{i}{1-b}\partial_x J_t^{(0)} + \left[\frac{H^{(0)} + bP^{(0)}}{1-b^2} + \frac{aQ^{(0)}}{1-b} + \frac{\bar{a}\bar{Q}^{(0)}}{1+b}, J_t^{(0)}\right]$$

$$= -\frac{i}{1-b}\partial_x J_t^{(0)} + \left(\frac{1}{1-b^2}\left[H^{(0)}, J_t^{(0)}\right] + \frac{ib}{1-b^2}\partial_x J_t^{(0)}\right) \qquad (3.9)$$

$$= \frac{1}{1-b^2}\left(\partial_x(-iJ_t^{(0)}) + \left[H^{(0)}, J_t^{(0)}\right]\right) = 0 \,,$$

216 where in the first line we plugged in the Ansatz, in the second we used the fact that
217 $[Q^{(0)}, J_t^{(0)}] = [\bar{Q}^{(0)}, J_t^{(0)}] = 0$ and that $[P, \mathcal{O}] = i\partial_x\mathcal{O}$, and in the third we discovered the
218 original conservation equation; recall that $J_x^{(0)} = -iJ_t^{(0)}$. Similarly, recycling the results
219 of the previous section and using that $[Q^{(0)}, J_t^{(0)}] = [\bar{Q}^{(0)}, J_t^{(0)}] = 0$, we learn that in the
220 stress tensor conservation equation we can focus on the linear in $a$ terms:

$$(\partial_x T_{tx} + [H, T_{tt}])\big|_{\text{linear in } a}$$

$$= \frac{i}{(1-b)^2}a\partial_x J_t^{(0)} + \left[\frac{aQ^{(0)}}{1-b}, \frac{1}{1-b^2}T_{tt}^{(0)} + \frac{ib}{1-b^2}T_{xt}^{(0)}\right] + \left[\frac{H^{(0)} + bP^{(0)}}{1-b^2}, -\frac{1}{1-b}aJ_t^{(0)}\right]$$

$$= ia\left(\frac{1}{(1-b)^2}\partial_x J_t^{(0)} + \frac{i}{(1-b^2)(1-b)}\left[H^{(0)}, J_t^{(0)}\right] + \frac{b}{(1-b^2)(1-b)}\partial_x J_t^{(0)}\right) = 0 \,,$$

$$(3.10)$$

221 where in the second equality we used that $[Q^{(0)}, T_{\mu\nu}^{(0)}] = 0$, and in the third that $\left[H^{(0)}, J_t^{(0)}\right] =$
222 $i\partial_x J_t^{(0)}$.

## 3.3 Ambiguities

223

224 There are ambiguities in the determination of currents from conservation laws. We could
225 perform an improvement transformation on the currents, $J_\mu = J_\mu^{(\text{min})} + \epsilon_{\mu\nu}\partial^\nu\chi$ with an
226 arbitrary scalar function $\chi$, which neither violates conservation nor changes the value of the
227 conserved charge. We also note that an improvement only changes the bilinear composite
228 operators $\mathcal{O} = \epsilon^{\mu\nu}J_\mu^{(1)}J_\mu^{(2)}$ by total derivatives, hence the theories deformed by $\mathcal{O}$ and
229 $\mathcal{O}^{(\text{min})}$ are equivalent. Another ambiguity arises from mixing two conserved currents.[1] In
230 the absence of Lorentz invariance mixing $J_\mu$ and $T_{\mu\nu}$ is allowed: an example that arises
231 in the discussion of Appendix D is the redefinition $\hat{J}_\mu \equiv J_\mu - 2\pi^2 i\ell\, T_{\bar{z}\mu}$. Finally, we could
232 simply multiply the conserved current by an arbitrary constant $\alpha$ to get a new conserved
233 current: $\hat{J}_\mu \equiv \alpha J_\mu$.

234  We have fixed all these ambiguities above by requiring that not just the charge $Q$,
235 but also the time component of the current $J_t$ remains unchanged. The only remaining
236 ambiguity that arises is that the spatial component of currents can be shifted by multiples
237 of the identity. The most general such transformations are:

$$T_{tt} = T_{tt}^{(\text{min})} + f_1(b)a^2 + 2f_2(b)a\bar{a} + f_3(b)\bar{a}^2 \,,$$
$$J_x = J_x^{(\text{min})} + 2i(f_1(b)a + f_2(b)\bar{a}) \,, \qquad \bar{J}_x = \bar{J}_x^{(\text{min})} - 2i(f_2(b)a + f_3(b)\bar{a}) \,,$$
$$T_{tx} = T_{tx}^{(\text{min})} - i\left(f_1'(b)a^2 + 2f_2'(b)a\bar{a} + f_3'(b)\bar{a}^2\right) \,,$$
$$T_{xx} = T_{tx}^{(\text{min})} + f_4(b)a^2 + 2f_5(b)a\bar{a} + f_6(b)\bar{a}^2 \,,$$

$$(3.11)$$

---

[1]In theories with a non-abelian symmetry group, the mixing ambiguity allows to change the $U(1)$
subgroup involved in our deformations. This ambiguity is fixed below together with all others.

where $\mathcal{O}^{(\mathrm{min})}$ refers to the expressions given in (3.7). These shifts satisfy dimensional constraints and the defining equations (3.8). We do not know of any algebraic way to fix these ambiguities. Using the Lagrangian formulation, however, we will fix these ambiguities in Section 4.3.

## 3.4   An attempt to deform by KdV currents

Encouraged by the success with turning on backgrounds for $T_{tx}$ and $J_x$, we attempt to deform by the higher spin KdV currents. For concreteness, we take the simplest one, obeying the conservation equation

$$\bar{\partial} T_4 = \partial \Theta_2 \,, \tag{3.12}$$

which in more convenient coordinates takes the form

$$
\begin{aligned}
0 &= \partial_x J_x^{(3)} + \partial_t J_t^{(3)} \,, \\
J_\mu^{(3)} &= \left( J_x^{(3)}, J_t^{(3)} \right) = \left( -2\pi i (T_4 - \Theta_2), 2\pi (T_4 + \Theta_2) \right) \,.
\end{aligned} \tag{3.13}
$$

We will use that $\partial_t J_t^{(3)} = [H, J_t^{(3)}]$ in the canonical formalism.

The corresponding conserved charge is

$$P_3 = \int dx \; J_t^{(3)}(x) \,. \tag{3.14}$$

We used the similarly defined conserved charges $P_{\pm 1}$ in (3.5). The KdV conserved charges are mutually commuting, $[P_s, P_\sigma] = 0$.

We want to introduce a background field $\alpha$ that couples to $J_x^{(3)}$, i.e.

$$\frac{\partial H(\alpha)}{\partial \alpha} = i \int dx \; J_x^{(3)} \,. \tag{3.15}$$

Specializing the argument of [2] to this case shows that all the $P_s$ can be preserved under this deformation: First, it is more convenient to work in the path integral formalism and define

$$P_s \equiv \frac{1}{2\pi} \oint_C \left( dz \, T_{s+1} + d\bar{z} \, \Theta_{s-1} \right) \,, \tag{3.16}$$

and then from $[P_s, P_\sigma] = 0$ it follows that

$$[P_s, T_{\sigma+1}(z)] = \partial A_{s,\sigma}(z) \,, \qquad [P_s, \Theta_{\sigma-1}(z)] = \bar{\partial} A_{s,\sigma}(z) \,. \tag{3.17}$$

Second, we assume that the theory at $\alpha$ has a conserved current $J_\mu^{(3)}$, and ask if we can adjust the current so that it remains conserved at $\alpha + \delta\alpha$, and that its charge commutes with the Hamiltonian. We now work out how $P_3$ changes under this deformation. We write:

$$
\begin{aligned}
0 &= \delta[H, P_3] = [\delta H, P_3] + [H, \delta P_3] \,, \\
[H, \delta P_3] &= i \int dx \; [P_3, J_x^{(3)}(x)] = -2\pi i \int dx \; \partial_t A_{3,3}(x) = \left[ H, -2\pi i \int dx \; A_{3,3}(x) \right] \,,
\end{aligned} \tag{3.18}
$$

where we used (3.13) and (3.17). From this we conclude that up to total derivatives

$$\delta J_t^{(3)}(x) = -2\pi i A_{3,3}(x) \,. \tag{3.19}$$

261 With a bit more work, it is possible to determine how to adjust $\delta J_x^{(3)}$ by local operators so
262 that that the current remains conserved.

263     Now we list some formulas valid at the CFT point. The KdV currents are well known,
264 and $A_{3,3}$ can be computed using the definition (3.17):

$$T_2 = T \qquad T_4 = :T^2:, \qquad T_6 = :T^3: + \frac{c+2}{12}:(\partial T)^2:, \qquad \dots$$

$$A_{3,3} = -4i:T^3: + \frac{i(c+2)}{2}:(\partial T)^2: + (\text{tot. der.}) = -4iT_6 + \frac{i5(c+2)}{6}:(\partial T)^2: + (\text{tot. der.}).$$
$$(3.20)$$

265 Using these formulas, for the family of theories defined by (3.15) to first order in $\alpha$ we get:

$$H = H^{(0)} + 2\pi\alpha P_3^{(0)} + O(\alpha^2),$$
$$P_3 = P_3^{(0)} - 8\pi\alpha \left( P_5^{(0)} - \frac{5(c+2)}{24} \int dx :(\partial T^{(0)})^2: \right) + O(\alpha^2),$$
$$(3.21)$$

266 where the (0) superscript indicates CFT quantities. Since $\int dx :(\partial T^{(0)})^2:$ does not commute
267 with $P_s^{(0)}$, unlike in the previously considered cases, we cannot use the CFT eigenstates
268 that simultaneously diagonalize $P_s^{(0)}$ as the eigenbasis for the deformed theories.[2] This in
269 itself does not constitute a no-go result, as in later sections we will be able to solve for
270 the spectrum of Hamiltonians that do not commute with $H^{(0)}$. However, we were not able
271 to understand how to extend (3.21) to all orders in $\alpha$ and how to obtain the spectrum of
272 (3.21) efficiently. It would be very interesting to make progress on these fronts.

# 4   Understanding the flow around a generic point

274 We saw that deformations by spatial components of the current and stress-tensor commute.
275 This is not true once we include the bilinear composite operators. Let us denote by
276 $\mathcal{H}(\lambda, a, \bar{a}, b)$ the Hamiltonian density that we obtain by first doing bilinear deformations,
277 *and then* turning on background gauge fields. See Figure 1 for a graphical representation.
278 We want to determine $\partial_\lambda \mathcal{H}(\lambda, a, \bar{a}, b)$ and solve the resulting equation using the initial
279 conditions determined in Section 3. The existence of such a universal equation valid for
280 all theories is already nontrivial, but the equation itself will have even more structure.
281 Schematically, we will find that for every bilinear operator (and their linear combinations)

$$\partial_\lambda \mathcal{H}(\lambda, a, \bar{a}, b) = g_1(b) \cdot \mathcal{O}_1 + a g_2(b) \cdot \mathcal{O}_2 + \bar{a} g_3(b) \cdot \mathcal{O}_3 + a^2 g_4(b) \cdot \mathcal{O}_4 + a\bar{a} g_5(b) \cdot \mathcal{O}_5 + \bar{a}^2 g_6(b) \cdot \mathcal{O}_6,$$
$$(4.1)$$

282 where we lightened the notation by introducing $g_I(b) \cdot \mathcal{O}_I = \sum_i g_{Ii}(b)\mathcal{O}_{Ii}$. This extra sum is
283 necessary, since there are different operators $\mathcal{O}_{Ii}$ multiplying a given power of $a$ and $\bar{a}$, with
284 different $b$ dependent coefficients. A remarkable property is that the RHS does not depend
285 on $\lambda$ explicitly. Since $[\lambda] \leq 0$ (the exact value depends on what operator we are deforming
286 by) and $[a, \bar{a}] = 1$, $[b] = 0$, positive powers of $\lambda$ would multiply high dimension operators (or
287 high powers of $a, \bar{a}$) on the RHS. The absence of $\lambda$ severely restricts the structure of the RHS:
288 we cannot have too high powers of $a, \bar{a}$ on dimensional grounds, and in practice $a, \bar{a}$ only
289 features quadratically. We note that there is another dimensionful quantity in the problem,

---

[2] We emphasize that $[H, P_3] = 0$ to all orders in $\alpha$, they just do not commute with $P_s^{(0)}$. To see that $[H, P_3] = 0$ to $O(\alpha^2)$ in (3.21), the only nontrivial step involves realizing that $\left[ H^0, \int dx :(\partial T^{(0)})^2: \right] = 0$, which is true because $:(\partial T^{(0)})^2:$ is holomorphic.

the length of the spatial $S^1$, $[L] = -1$. Since we are (at least formally) working in local field theory, it cannot appear in (4.1). In order for (4.1) to be unambiguous, we need the operators $\mathcal{O}_{Ii}$ to be either $J_\mu$, $\bar{J}_\mu$, $T_{\mu\nu}$ or one of the factorizing bilinear composite operators built from them, as generic composite operators have arbitrariness in their definitions. The $\mathcal{O}_{Ii}$ we find are indeed such special operators.

If there exists a universal equation for deformations by higher spin KdV currents similar to (4.1), we expect that it would involve an infinite number of terms on the RHS. This has simple dimensional reasons. The background fields coupling to the spin-$s$ higher spin currents have dimension $[\alpha_s] = 2 - s$, and hence for $s > 2$ it is an irrelevant coupling. We expect that arbitrary high powers of it would appear on the RHS with very irrelevant factorizing composite operators built from the KdV currents multiplying them. It would be interesting to understand if a universal equation exists at all, and whether our expectations about it are realized.

Once we have the operator equation (4.1), we can take the diagonal matrix element in the eigenstate $|n\rangle$, use the Hellmann-Feynman theorem on the LHS as in (2.3), factorization on the RHS for composite operators, and compute the matrix elements according to what was explained in Section 2. This then leads to a flow equation for the energy eigenvalues in the enlarged coupling space.

We are not able to derive (4.1) in a systematic manner. We will find the equation for deformations starting from the classical free scalar theory nonperturbatively in Section 4.1. We then check the validity of the equation in a more general classical field theory in Section 5.1, and in the quantum theory at low orders in perturbation theory in Section 5.2. We solve the equations in Section 6.1. The solution reproduces the energy spectrum of the $T\bar{T}$ and $J\bar{T}$ deformed theories obtained previously in the literature as special cases. We also compute the spectrum of a certain string theory in Section 6.4, which was argued in [5] to be dual to a theory that is closely related to a CFT deformed by $T\bar{T}$, $J\bar{T}$, $\bar{J}T$ simultaneously; the results are again in perfect agreement.

## 4.1 Flow equation for the classical free scalar

It is clear that if a universal operator equation like (4.1) exists, then it must hold for classical field theories. Conversely, we can use classical field theory to conjecture the equation (4.1), and then test it in the quantum theory. There also exists a way to read off the energy levels from the knowledge of the classical Hamiltonian, assuming that a universal expression for these also exists, see Appendix C.

To keep the discussion simple, we will study the case of the $J\bar{T}$ deformation of the free massless scalar first. To reiterate, we want determine the Hamiltonian density $\mathcal{H}(\lambda, a, b)$ by first deforming by $\lambda J\bar{T}$, and then by $aJ_x$ and by $bT_{tx}$. This implies that $\partial_\lambda \mathcal{H}(\lambda, a, b)$ is not just $J\bar{T}$. As we will see it is instead a linear combination of various deformations, see also Figure 1. For our conventions for the free scalar see Appendix B, this helps explain some signs that appear below.

Now take $\mathcal{H} = h(\partial_x \phi, \Pi)$. We enforce the quantization of charges and momentum by requiring that the $t$ components of the corresponding currents do not depend on the

couplings $\lambda, a, b$ and we obtain their $x$ components from conservation:

$$J_t = -\frac{1}{2}(\partial_x\phi - 4\pi\Pi)\,, \qquad J_x = 2\pi i\left(\frac{\partial h}{\partial(\partial_x\phi)} - \frac{1}{4\pi}\frac{\partial h}{\partial\Pi}\right),$$

$$\bar{J}_t = -\frac{1}{2}(\partial_x\phi + 4\pi\Pi)\,, \qquad \bar{J}_x = -2\pi i\left(\frac{\partial h}{\partial(\partial_x\phi)} + \frac{1}{4\pi}\frac{\partial h}{\partial\Pi}\right),$$

$$T_{tt} = -h\,, \qquad\qquad T_{tx} = -i\frac{\partial h}{\partial\Pi}\frac{\partial h}{\partial(\partial_x\phi)}\,, \tag{4.2}$$

$$T_{xt} = -i\,\Pi\partial_x\phi\,, \qquad T_{xx} = \Pi\frac{\partial h}{\partial\Pi} + \partial_x\phi\frac{\partial h}{\partial(\partial_x\phi)} - h\,.$$

It is easy to check that the currents are conserved using Hamilton's equations (B.6), and they reduce to their free scalar counterparts listed in Appendix B.

Deforming $\mathcal{H}$ by $J_x$ or $\bar{J}_x$ amounts to shifting $\partial_x\phi$ and $\Pi$. In particular, $\mathcal{H}(a) = h(\partial_x\phi - 2\pi a, \Pi + a/2)$ obeys

$$\frac{\partial\mathcal{H}(a)}{\partial a} = iJ_x(a)\,, \tag{4.3}$$

where $J_x(a)$ is the spatial component of the current in the presence of the background gauge field $a$. The deformation of $\mathcal{H}$ by $ibT_{tx}$ cannot be written in a closed form in general,

$$\mathcal{H}(b) = h - b\frac{\partial h}{\partial(\partial_x\phi)}\frac{\partial h}{\partial\Pi} + O(b^2)\,, \quad \text{so that} \quad \frac{\partial\mathcal{H}(b)}{\partial b} = -iT_{tx}(b)\,. \tag{4.4}$$

After this preparation, consider any deformation with the background fields $a, b$ set to zero:

$$\partial_\lambda\mathcal{H}(\lambda) = S\left(\mathcal{H}, \frac{\partial\mathcal{H}}{\partial(\partial_x\phi)}, \frac{\partial\mathcal{H}}{\partial\Pi}, \partial_x\phi, \Pi\right) \tag{4.5}$$

for some $S$ that is a function of its five argument. Note that all the components of the currents in (4.2) are of this form, hence the deformations (background gauge fields and bilinear deformations) of interest in this paper are a special case of $S$. Let $\mathcal{H}(\lambda, a, b)$ be obtained by turning on background fields $a$ and $b$ *after* deforming by $\lambda$. Below, in (4.14), we give an explicit formula for $\partial_\lambda\mathcal{H}(\lambda, a, b)$ in terms of $S$, which is the main result of this section.

Let us define

$$\tilde{S}(a, b, \partial_x\phi, \Pi) \equiv S\left(\mathcal{H}(\lambda, a, b), \frac{\partial\mathcal{H}(\lambda, a, b)}{\partial(\partial_x\phi)}, \frac{\partial\mathcal{H}(\lambda, a, b)}{\partial\Pi}, \partial_x\phi, \Pi\right)\,, \tag{4.6}$$

i.e. $\tilde{S}$ is the same function as $S$, but regarded as having the arguments $(a, b, \partial_x\phi, \Pi)$. As a first step towards obtaining $\partial_\lambda\mathcal{H}(\lambda, a, b)$, let us set $b = 0$. Then from what we said around (4.3) it follows that

$$\partial_\lambda\mathcal{H}(\lambda, a, 0, \partial_x\phi, \Pi) = \tilde{S}(0, 0, \partial_x\phi - 2\pi a, \Pi + a/2)\,. \tag{4.7}$$

As discussed around (4.4), obtaining such a closed form formula for $b \neq 0$ does not seem possible, but perturbation theory should be straightforward. Taking this as a hint, we expand the RHS of (4.7) in $a$:[3]

$$\tilde{S}(0, 0, \partial_x\phi - 2\pi a, \Pi + a/2) = \sum_{m\geq 0}\frac{a^m}{m!}D_1^m\tilde{S}(a, 0, \partial_x\phi, \Pi)\,, \tag{4.8}$$

---

[3]The point is to express $\partial_\lambda\mathcal{H}$ in terms of the currents and their bilinears at the same value of $a$ instead of $a = 0$.

353  where the differential operator $D_1$ is defined by:

$$D_1 = -2\pi \left( \frac{\partial}{\partial(\partial_x\phi)} - \frac{1}{4\pi} \frac{\partial}{\partial\Pi} \right) - \partial_a \,. \tag{4.9}$$

354  The infinite series is easily seen to implement a translation $(-a, 0, 2\pi a, -a/2)$ on the
355  arguments of $\tilde{S}$, proving (4.8). Another nice way to see that the infinite series is equal to
356  $\partial_\lambda \mathcal{H}(\lambda, a, 0)$, is to prove that they satisfy the same differential equation (regarded as an
357  evolution equation with $a$ as time) with the same initial condition:

$$D_1 \partial_\lambda \mathcal{H}(\lambda, a, 0, \partial_x\phi, \Pi) = \partial_\lambda \left( D_1 \mathcal{H}(\lambda, a, 0, \partial_x\phi, \Pi) \right) = 0 \,,$$

$$D_1 \sum_{m\geq 0} \frac{a^m}{m!} D_1^m \tilde{S}(a, 0, \partial_x\phi, \Pi) = \left( -\sum_{m\geq 1} \frac{a^{m-1}}{(m-1)!} D_1^m + \sum_{m\geq 0} \frac{a^m}{m!} D_1^{m+1} \right) \tilde{S}(a, 0, \partial_x\phi, \Pi) = 0 \,,$$

$$\partial_\lambda \mathcal{H}(\lambda, 0, 0, \partial_x\phi, \Pi) = \tilde{S}(0, 0, \partial_x\phi, \Pi) \,,$$

$$\tag{4.10}$$

358  where in the first line we used that $[D_1, \partial_\lambda] = 0$ (because partial derivatives commute) and
359  (4.3) together with the expression of $J_x$ given in (4.2), while in the second line we relabeled
360  the summation index to show that the two terms cancel. The third line is true by the
361  definition of the $\lambda$ deformation (4.5).
362      The latter method generalizes to the $b \neq 0$ case. We want to find a differential operator
363  $D_2 = -\partial_b + \ldots$ that annihilates $\partial_\lambda \mathcal{H}$. The unique $D_2$ satisfying this regardless of $\partial_\lambda \mathcal{H}$ is:

$$D_2 = -\left( \frac{\partial\mathcal{H}(\lambda, a, b, \partial_x\phi, \Pi)}{\partial(\partial_x\phi)} \frac{\partial}{\partial\Pi} + \frac{\partial\mathcal{H}(\lambda, a, b, \partial_x\phi, \Pi)}{\partial\Pi} \frac{\partial}{\partial(\partial_x\phi)} \right) - \partial_b \,. \tag{4.11}$$

364  To show that $D_2 \partial_\lambda \mathcal{H} = 0$ we have to do some computations, as unlike in (4.10), $[D_2, \partial_\lambda] \neq 0$
365  and $D_2 \mathcal{H} \neq 0$. We write

$$\begin{aligned} D_2 \partial_\lambda \mathcal{H} &= -\left( \partial_b \partial_\lambda \mathcal{H} + \frac{\partial\mathcal{H}}{\partial(\partial_x\phi)} \frac{\partial(\partial_\lambda\mathcal{H})}{\partial\Pi} + \frac{\partial\mathcal{H}}{\partial\Pi} \frac{\partial(\partial_\lambda\mathcal{H})}{\partial(\partial_x\phi)} \right) \\ &= -\partial_\lambda \left( \partial_b \mathcal{H} + \frac{\partial\mathcal{H}}{\partial(\partial_x\phi)} \frac{\partial\mathcal{H}}{\partial\Pi} \right) = 0 \,, \end{aligned} \tag{4.12}$$

366  where in the first line we wrote out the definitions, and in the second we commuted partial
367  derivatives, and used that $\mathcal{H}$ satisfies the differential equation displayed in (4.4). Then we
368  follow the same logic as in (4.10). The proof of

$$D_2 \sum_{n\geq 0} \frac{b^n}{n!} D_2^n \tilde{S}(0, b, \partial_x\phi, \Pi) = 0 \tag{4.13}$$

369  follows that in (4.10) verbatim. Thus the series and $\partial_\lambda \mathcal{H}$ satisfy the same evolution equation
370  in $b$ (regarded as time), with the same initial conditions, given in the last line of (4.10).
371  It is easy to show that $[D_1, D_2] = 0$, hence we can combine the two evolutions without
372  encountering any issues, and we arrive at

$$\boxed{\partial_\lambda \mathcal{H}(\lambda, a, b, \partial_x\phi, \Pi) = \sum_{m,n\geq 0} \frac{1}{m!n!} a^m b^n D_1^m D_2^n \tilde{S}(a, b, \partial_x\phi, \Pi) \,.} \tag{4.14}$$

373  This is our key result, we will see that the infinite sum truncates in the cases of interest,
374  and the resulting equation will allow us to write down an evolution equation for the energy
375  levels.

To be able to deform by operators built from $\bar{J}$, we introduce a background field $\bar{a}$ that couples to it. As defined in (3.8), the analogue of (4.3) is $\frac{\partial \mathcal{H}}{\partial \bar{a}} = -i\bar{J}_x$. The corresponding unique differential operator that annihilates $\partial_\lambda \mathcal{H}(\lambda, a, \bar{a}, b)$ is given by:

$$\bar{D}_1 = -2\pi \left( \frac{\partial}{\partial(\partial_x \phi)} + \frac{1}{4\pi} \frac{\partial}{\partial \Pi} \right) - \partial_{\bar{a}} \,. \tag{4.15}$$

$\bar{D}_1$ commutes with $D_1$, $D_2$.

Because we intend to build the deforming operator $S$ from $T_{\mu\nu}$, $J_\mu$, $\bar{J}_\mu$, it is a useful intermediate step to compute the action of the differential operators on these quantities. Remarkably, this results in components of conserved currents. We collect the results in Table 1.

|  | $J_t$ | $J_x$ | $\bar{J}_t$ | $\bar{J}_x$ | $T_{tt}$ | $T_{tx}$ | $T_{xt}$ | $T_{xx}$ |
|---|---|---|---|---|---|---|---|---|
| $D_1$ | $2\pi$ | $0$ | $0$ | $0$ | $0$ | $0$ | $iJ_t$ | $iJ_x$ |
| $\bar{D}_1$ | $0$ | $0$ | $2\pi$ | $0$ | $0$ | $0$ | $-i\bar{J}_t$ | $-i\bar{J}_x$ |
| $D_2$ | $iJ_x$ | $0$ | $i\bar{J}_x$ | $0$ | $iT_{tx}$ | $0$ | $-i(T_{tt} - T_{xx})$ | $-iT_{tx}$ |

Table 1: Action of the differential operators $D_1$, $\bar{D}_1$, $D_2$ on the operators $J_\mu$, $\bar{J}_\mu$, $T_{\mu\nu}$.

As promised, we go through the $J\bar{T}$ deformation in detail, and to shorten the discussion, we set $\bar{a} = 0$. We will later write down the result including $\bar{a} \neq 0$ terms and also for the rest of the bilinear deformations. By the $J\bar{T}$ deformation, we mean that we add to the Hamiltonian

$$\begin{aligned} S_{J\bar{T}} &= 2\pi i J_{[t|} T_{\bar{z}|x]} \\ &= \frac{\pi i}{2} \left( J_t T_{xx} + i J_t T_{tx} - J_x T_{xt} - i J_x T_{tt} \right) \,, \end{aligned} \tag{4.16}$$

where the normalization is chosen so that we get $S_{J\bar{T}} = J\bar{T}$ in a CFT; our conventions are summarized in Appendix A. We then compute all the non-vanishing derivatives (omitting $\bar{D}_1$):

$$\begin{aligned} D_1 S_{J\bar{T}} &= 2\pi^2 i T_{\bar{z}x} & D_2 S_{J\bar{T}} &= \pi J_{[t|} T_{t|x]} \\ D_1^2 S_{J\bar{T}} &= -\pi^2 J_x & D_1 D_2 S_{J\bar{T}} &= \pi^2 T_{tx} \,. \end{aligned} \tag{4.17}$$

All other derivatives vanish. Using these formulas we conclude that

$$\partial_\lambda \mathcal{H}(\lambda, a, b) = 2\pi i J_{[t|} T_{\bar{z}|x]} + \pi b J_{[t|} T_{t|x]} - \frac{\pi^2 a^2}{2} J_x + 2\pi^2 i a T_{\bar{z}x} + \pi^2 ab T_{tx} \,. \tag{4.18}$$

This universal equation holds for any Hamiltonian density in the class we considered. The term $\pi b J_{[t|} T_{t|x]}$ is a linear combination of $J_z \bar{T} + J_{\bar{z}} \bar{\Theta}$ and $J_z \Theta + J_{\bar{z}} T$ deformations. In a CFT the second deformation vanishes.

Now we are ready to systematize the derivation for all bilinear deformations that we can construct. The bilinear composite operators that obey factorization are

$$\begin{aligned} \text{``}J\bar{J}\text{''} &\equiv -i J_{[t} \bar{J}_{x]} \\ \text{``}J\bar{T}\text{''} &\equiv 2\pi i J_{[t|} T_{\bar{z}|x]} \\ \text{``}J\Theta\text{''} &\equiv -2\pi i J_{[t|} T_{z|x]} \\ \text{``}\bar{J}T\text{''} &\equiv -2\pi i \bar{J}_{[t|} T_{z|x]} \\ \text{``}\bar{J}\bar{\Theta}\text{''} &\equiv 2\pi i \bar{J}_{[t|} T_{\bar{z}|x]} \\ \text{``}T\bar{T}\text{''} &\equiv -2\pi^2 T_{t[t|} T_{x|x]} \,. \end{aligned} \tag{4.19}$$

Instead of writing six long equations, we give $\frac{\partial}{\partial\lambda_{\mathcal{O}}}\mathcal{H}(\lambda, a, \bar{a}, b)$ with the deforming composite operator being $\mathcal{O}$ in Table 2. As promised, the equation is of the form (4.1).

| $\mathcal{O}$ ＼ $+$ | $J\bar{J}$ | $J\bar{T}$ | $J\Theta$ | $\bar{J}T$ | $\bar{J}\bar{\Theta}$ | $T\bar{T}$ | $J_t$ | $J_x$ | $\bar{J}_t$ | $\bar{J}_x$ | $T_{tt}$ | $T_{tx}$ | $T_{xt}$ | $T_{xx}$ |
|---|---|---|---|---|---|---|---|---|---|---|---|---|---|---|
| $J\bar{J}$ | 1 | 0 | 0 | 0 | 0 | 0 | 0 | $i\pi\bar{a}$ | 0 | $-i\pi a$ | 0 | 0 | 0 | 0 |
| $J\bar{T}$ | $i\pi\bar{a}$ | $1-\frac{b}{2}$ | $-\frac{b}{2}$ | 0 | 0 | 0 | 0 | $-\frac{\pi^2}{2}\left(a^2+\bar{a}^2\right)$ | 0 | $\pi^2 a\bar{a}$ | 0 | $-\pi^2 a(1-b)$ | 0 | $i\pi^2 a$ |
| $J\Theta$ | $-i\pi\bar{a}$ | $\frac{b}{2}$ | $1+\frac{b}{2}$ | 0 | 0 | 0 | 0 | $\frac{\pi^2}{2}\left(a^2+\bar{a}^2\right)$ | 0 | $-\pi^2 a\bar{a}$ | 0 | $-\pi^2 a(1+b)$ | 0 | $-i\pi^2 a$ |
| $\bar{J}T$ | $-i\pi a$ | 0 | 0 | $1+\frac{b}{2}$ | $\frac{b}{2}$ | 0 | 0 | $\pi^2 a\bar{a}$ | 0 | $-\frac{\pi^2}{2}\left(a^2+\bar{a}^2\right)$ | 0 | $-\pi^2\bar{a}(1+b)$ | 0 | $-i\pi^2\bar{a}$ |
| $\bar{J}\bar{\Theta}$ | $i\pi a$ | 0 | 0 | $-\frac{b}{2}$ | $1-\frac{b}{2}$ | 0 | 0 | $-\pi^2 a\bar{a}$ | 0 | $\frac{\pi^2}{2}\left(a^2+\bar{a}^2\right)$ | 0 | $-\pi^2\bar{a}(1-b)$ | 0 | $i\pi^2\bar{a}$ |
| $T\bar{T}$ | 0 | $-i\pi a$ | $-i\pi a$ | $i\pi\bar{a}$ | $i\pi\bar{a}$ | 1 | 0 | 0 | 0 | 0 | 0 | $i\pi^3\left(a^2-\bar{a}^2\right)$ | 0 | 0 |

Table 2: The equation for $\frac{\partial}{\partial\lambda_{\mathcal{O}}}\mathcal{H}(\lambda, a, \bar{a}, b)$ can be read out from this table as follows. The deforming operator $S = \mathcal{O}$ labels the rows. We have to add up the operators in the top row with coefficients in the row labelled by $\mathcal{O}$. For comparison, the $J\bar{T}$ example for $\bar{a} = 0$ is given in (4.18) in more conventional form.

We postpone solving these equations. Instead, we convert them now into equations describing the evolution of the spectrum. In Appendix C we then explain how to recover the classical Hamiltonian (and Lagrangian) from the solution of the spectrum.

## 4.2 Flow equation for the spectrum

The flow equations for the Hamiltonian density, (4.18) and Table 2, can now be turned into a flow equation for the energy eigenvalues following the strategy outlined in Section 2: for a given eigenstate $|n\rangle$, we take the diagonal matrix element of the (conjectured) operator equation, for the composite operators use factorization, and replace the matrix elements that we encounter with:

$$\langle n|\frac{\partial}{\partial\lambda_{\mathcal{O}}}\mathcal{H}(\lambda, a, \bar{a}, b)|n\rangle = \frac{\partial}{\partial\lambda_{\mathcal{O}}}E_n(\lambda, a, \bar{a}, b),$$

$$\langle n|J_t|n\rangle = \frac{Q_n}{L}, \qquad \langle n|J_x|n\rangle = -\frac{i\partial_a E_n}{L}, \qquad \langle n|\bar{J}_t|n\rangle = \frac{\bar{Q}_n}{L}, \qquad \langle n|\bar{J}_x|n\rangle = \frac{i\partial_{\bar{a}} E_n}{L}$$

$$\langle n|T_{tt}|n\rangle = -\frac{E_n}{L}, \quad \langle n|T_{tx}|n\rangle = \frac{i\partial_b E_n}{L}, \qquad \langle n|T_{xt}|n\rangle = \frac{iP_n}{L}, \quad \langle n|T_{xx}|n\rangle = -\partial_L E_n.$$

$$(4.20)$$

For the time component of currents, $J_t$, $\bar{J}_t$, $T_{tt}$, $T_{xt}$ the above equations follow from the definition of charge given in (A.4) and (A.10). We coupled the spatial components of the currents to background fields, see (4.3) and (4.4), thereby modifying the Hamiltonian, and we use the Hellmann-Feynman theorem $\langle n|\partial_\lambda H|n\rangle = \partial_\lambda E_n$ to determine their matrix elements. The same logic is used to determine the first line of (4.20). The matrix element of $T_{xx}$ is curious, we obtain $-\partial_L E_n$ from its interpretation as pressure. From our perspective, the length of the spatial $S^1$ can be regarded as a *background field* on the same footing as $a, \bar{a}, b$, and from this point of view it becomes natural that its diagonal matrix element is obtained by taking a $\partial_L$ derivative.

Executing this straightforward, but tedious task, we arrive at the differential equation describing the flow of energy eigenvalues. We again put the equations in a table, see Table 3.

419    For ease of reading, we write out the equation for the $J\bar{T}$ deformation explicitly:

$$0 = \frac{2L}{i\pi}\frac{\partial}{\partial\lambda_{J\bar{T}}}E_n + \left(-\bar{a}\hat{\bar{Q}}_n - \pi(a^2-\bar{a}^2)L - (1-b)E_n + P_n\right)\partial_a E_n$$
$$-\bar{a}\hat{Q}_n\partial_{\bar{a}}E_n + (1-b)\hat{Q}_n\partial_b E_n + L\hat{Q}_n\partial_L E_n\,, \tag{4.21}$$
$$\hat{Q} \equiv Q + 2\pi aL\,, \qquad \hat{\bar{Q}} \equiv \bar{Q} + 2\pi\bar{a}L\,.$$

| $\mathcal{O}$ $\diagdown$ $+$ | $\frac{\partial}{\partial\lambda_{\mathcal{O}}}E_n$ | $\partial_a E_n$ | $\partial_{\bar{a}}E_n$ | $\partial_b E_n$ | $\partial_L E_n$ |
|---|---|---|---|---|---|
| $J\bar{J}$ | $2L$ | $-\hat{\bar{Q}}_n$ | $-\hat{Q}_n$ | $0$ | $0$ |
| $J\bar{T}$ | $\frac{2L}{i\pi}$ | $-\bar{a}\hat{\bar{Q}}_n - \pi(a^2-\bar{a}^2)L$ $-(1-b)E_n + P_n$ | $-\bar{a}\hat{Q}_n$ | $(1-b)\hat{Q}_n$ | $L\hat{Q}_n$ |
| $J\Theta$ | $\frac{2L}{i\pi}$ | $\bar{a}\hat{\bar{Q}}_n + \pi(a^2-\bar{a}^2)L$ $-(1+b)E_n - P_n$ | $\bar{a}\hat{Q}_n$ | $(1+b)\hat{Q}_n$ | $-L\hat{Q}_n$ |
| $\bar{J}T$ | $-\frac{2L}{i\pi}$ | $-a\hat{\bar{Q}}_n$ | $-a\hat{Q}_n + \pi(a^2-\bar{a}^2)L$ $-(1+b)E_n - P_n$ | $-(1+b)\hat{\bar{Q}}_n$ | $L\hat{\bar{Q}}_n$ |
| $\bar{J}\bar{\Theta}$ | $-\frac{2L}{i\pi}$ | $a\hat{\bar{Q}}_n$ | $a\hat{Q}_n - \pi(a^2-\bar{a}^2)L$ $-(1-b)E_n + P_n$ | $-(1-b)\hat{\bar{Q}}_n$ | $-L\hat{\bar{Q}}_n$ |
| $T\bar{T}$ | $-\frac{L}{\pi^2}$ | $aE_n$ | $\bar{a}E_n$ | $-a\hat{Q}_n + \bar{a}\hat{\bar{Q}}_n$ $\pi(a^2-\bar{a}^2)L - P_n$ | $-E_nL$ |

Table 3: The flow equation for the energy eigenvalue can be read out from the this table as follows. The deforming operator $S = \mathcal{O}$ labels the rows. We have to add up the terms in the top row with coefficients in the row labelled by $\mathcal{O}$ and equate it to zero. For reference, the second line is given in conventional form in (4.21), where we also define $\hat{Q}, \hat{\bar{Q}}$.

420

421    The main power of our method comes from its ability to solve theories where we consider
422 a linear combination of irrelevant deformations. Recall that, as reviewed in Section 2,
423 the case of $T\bar{T}$ deformation of a relativistic QFT can be solved without introducing the
424 background fields $a, \bar{a}, b$ [2,3], while the $J\bar{T}$ (or equivalently the $\bar{J}T$) deformation can be
425 solved using holomorphy [5]. However, the combination of $T\bar{T}$ and $J\bar{T}$ leads to the loss of
426 both Lorentz invariance and holomorphy, and the aforementioned methods do not apply.
427    Let us introduce a length scale $\ell$ with $[\ell] = -1$ and real dimensionless couplings $g_{\mathcal{O}}$:

$$\lambda_{J\bar{T}} \equiv ig_{J\bar{T}}\ell\,, \qquad \lambda_{J\Theta} \equiv ig_{J\Theta}\ell\,, \qquad \lambda_{\bar{J}T} \equiv -ig_{\bar{J}T}\ell\,, \qquad \lambda_{\bar{J}\bar{\Theta}} \equiv -ig_{\bar{J}\bar{\Theta}}\ell\,,$$
$$\lambda_{T\bar{T}} \equiv g_{T\bar{T}}\ell^2\,. \tag{4.22}$$

428 By changing $\ell$, we obtain a one-parameter family of theories. Note that because $J\bar{J}$ is a
429 marginal operator it is not included among the deforming operators. The energy levels
430 evolve according to the equation:

$$L\frac{\partial}{\partial\ell}E_n = \frac{\pi g_{J\bar{T}}}{2}\mathrm{II} + \frac{\pi g_{J\Theta}}{2}\mathrm{III} + \frac{\pi g_{\bar{J}T}}{2}\mathrm{IV} + \frac{\pi g_{\bar{J}\bar{\Theta}}}{2}\mathrm{V} + 2\pi^2\ell\, g_{T\bar{T}}\mathrm{VI}\,, \tag{4.23}$$

431 where the Roman numerals stand for one row of Table 3 (omitting the $\frac{\partial}{\partial\lambda_{\mathcal{O}}}E_n$ entry). We
432 note that a similar equation can also be obtained at the level of the operator equations
433 included in Table 2. We will solve (4.23) in Section 6.1 with the initial conditions determined
434 in Section 3.

### 4.3 Fixing ambiguities in the initial conditions

In Section 3.3, we discussed some ambiguities in the initial conditions. These ambiguities are fixed by the form of the conserved currents that we gave in (4.2). Conversely, the $x$ components of currents in (4.2) could be shifted in the same way as in (3.11) while preserving conservation. Since we have the additional scale $\ell$ with $[\ell] = -1$ in the problem, the ambiguities could be made even more severe than those in the initial conditions. We have to invoke additional principles to fix them.

Let us start with $T_{xx}$, from which we want to require $\langle n|T_{xx}|n\rangle = -\partial_L E_n$, see (4.20). The Noether stress tensor given in (4.2) achieves this. Since we have not written down $T_{xx}^{(\mathrm{min})}$ there, we omit the details and just state that there indeed exists a shift involving the background fields that makes the expression of $T_{xx}$ in (4.20) match with $T_{xx}^{(\mathrm{min})}$.

Coupling a scalar theory to a constant background gauge field $A_\mu = (a, 0)$ in the Hamiltonian formalism amounts to the shift $\mathcal{H}(a) = h(\partial_x \phi - 2\pi a, \Pi + a/2)$. This was already used above, see (4.3). Gauge invariance forbids the addition of $A_\mu A^\mu$ terms. At $\lambda_{\mathcal{O}} = \ell = 0$, we determined the Hamiltonian in Appendix B, in (B.10). Comparing this to the algebraic result $T_{tt}^{(\mathrm{min})}$, we require the shift:

$$T_{tt} = T_{tt}^{(\mathrm{min})} - \frac{\pi a^2}{1-b} - \frac{\pi \bar{a}^2}{1+b},\tag{4.24}$$

and the shifts of of $J_x^{(\mathrm{min})}$, $\bar{J}_x^{(\mathrm{min})}$, $T_{tx}^{(\mathrm{min})}$ follow from these shifts according to (3.11). We have checked that these shifts are exactly the ones needed to reproduce the currents given in (4.2). Integrating (4.24) according to the rule (A.4), we get

$$H(a, \bar{a}, b) = \frac{H^{(0)} + bP^{(0)}}{1-b^2} + \frac{aQ^{(0)} + \pi a^2 L}{1-b} + \frac{\bar{a}\bar{Q}^{(0)} + \pi \bar{a}^2 L}{1+b},\tag{4.25}$$

where we used (3.6).

After settling the ambiguities, we are ready to give the initial conditions for the energy flow equations. Because the operators in (4.25) commute, we can easily convert it to an expression for the energy eigenvalues:

$$E_n = \langle n|H(a, \bar{a}, b)|n\rangle = \frac{E_n^{(0)} + bP_n}{1-b^2} + \frac{aQ_n + \pi a^2 L}{1-b} + \frac{\bar{a}\bar{Q}_n + \pi \bar{a}^2 L}{1+b}.\tag{4.26}$$

We will use this as initial data for the flow equation (4.23) in Section 6.1.

We remark that the algebraic approach does not break down without the additional requirements discussed in this section. E.g. we could define $J_x = 2\pi i \left( \frac{\partial \mathcal{H}}{\partial(\partial_x \phi)} - \frac{1}{4\pi} \frac{\partial \mathcal{H}}{\partial \Pi} \right) + 2iag_1(b)$, which would in turn lead to the modification of entries in Tables 1, 2, 3, and ultimately lead to a different (and uglier) (4.23). The solution would also change, but setting the background fields to zero must give an identical result for the energy spectrum of the theory deformed by bilinear composite operators.

## 5 Checks

### 5.1 A classical field theory check

In the previous section we conjectured a set of universal equations, (4.18) and Table 2, governing the evolution of the Hamiltonian under irrelevant deformations based on the classical free scalar with shift symmetry. In this section, we check a restriction of these

equations to the case of one conserved $U(1)$ current which can generate a symmetry other than shifts, and a much more general classical scalar theory.

Consider a collection of scalars $\phi_I$ and momenta $\Pi^I$ and Hamiltonian density $H = h(\phi_I, \partial_x \phi_I, \Pi^I)$, for example scalars with a potential or a sigma model. We sum over repeated $I, J, \dots$ indices. The Hamilton equations of motion are:

$$\partial_t \phi_I = -i \frac{\partial \mathcal{H}}{\partial \Pi^I}, \qquad \partial_t \Pi^I = i \left( \frac{\partial \mathcal{H}}{\partial \phi_I} - \partial_x \frac{\partial \mathcal{H}}{\partial (\partial_x \phi_I)} \right). \tag{5.1}$$

For our sign conventions refer to (B.6). The theory is translation invariant and in complete analogy to (4.2) the components of the conserved stress tensor are:

$$\begin{aligned} T_{tt} &= -\mathcal{H}, \qquad T_{tx} = -i \frac{\partial \mathcal{H}}{\partial \Pi^I} \frac{\partial \mathcal{H}}{\partial (\partial_x \phi_I)}, \\ T_{xt} &= -i \Pi^I \partial_x \phi_I, \qquad T_{xx} = \frac{\partial \mathcal{H}}{\partial (\partial_x \phi_I)} \partial_x \phi_I + \frac{\partial \mathcal{H}}{\partial \Pi^I} \Pi^I - \mathcal{H}. \end{aligned} \tag{5.2}$$

If the Hamiltonian is invariant under some continuous symmetry group acting like $\delta \phi_I = \Lambda_I(\phi)$ and $\delta \Pi^I = -\Pi^J \frac{\partial \Lambda_J}{\partial \phi_I}$, it has a conserved current

$$K_t = \Pi^I \Lambda_I, \qquad K_x = i \frac{\partial \mathcal{H}}{\partial (\partial_x \phi_I)} \Lambda_I. \tag{5.3}$$

For the familiar case of the $O(2)$ symmetric scalar field, we have $\Lambda_I = \epsilon_{IJ} \phi_J$. For the shift symmetry we discussed in Section 4.1, $\Lambda = 4\pi$ and the current $K$ of (5.3) corresponds to the difference of holomorphic and antiholomorphic currents $K_\mu = J_\mu - \bar{J}_\mu$, hence we chose a different name for it.

We want to understand deformations by coupling to the background fields $a$ and $b$ according to the rules $\frac{\partial \mathcal{H}}{\partial a} = i K_x$, $\frac{\partial \mathcal{H}}{\partial b} = -i T_{tx}$. Following the strategy of Section 4.1 to write down a flow equation for $\mathcal{H}(\lambda, a, b)$, we want to find commuting differential operators $\mathcal{D}_1, \mathcal{D}_2$ that act on functions of variables $(a, b, \phi_I, \partial_x \phi_I, \Pi^I)$ and that annihilate $\partial_\lambda \mathcal{H}$. This is possible, and their expressions are:

$$\begin{aligned} \mathcal{D}_1 &= -\Lambda_I \frac{\partial}{\partial (\partial_x \phi_I)} - \partial_a, \\ \mathcal{D}_2 &= -\left( \frac{\partial \mathcal{H}}{\partial (\partial_x \phi_I)} \frac{\partial}{\partial \Pi^I} + \frac{\partial \mathcal{H}}{\partial \Pi^I} \frac{\partial}{\partial (\partial_x \phi_I)} \right) - \partial_b. \end{aligned} \tag{5.4}$$

It is now straightforward to compute the results in Table 4. Note that in the case of the scalar with shift symmetry investigated in Section 4.1, $\bar{a} = a$ and $\mathcal{D}_1 = D_1 + \bar{D}_1$. The results in this table are in complete agreement with those in Table 1, if we remember that $K_\mu = J_\mu - \bar{J}_\mu$ and $\mathcal{D}_1 = D_1 + \bar{D}_1$.

|  | $K_t$ | $K_x$ | $T_{tt}$ | $T_{tx}$ | $T_{xt}$ | $T_{xx}$ |
|---|---|---|---|---|---|---|
| $\mathcal{D}_1$ | 0 | 0 | 0 | 0 | $iK_t$ | $iK_x$ |
| $\mathcal{D}_2$ | $iK_x$ | 0 | $iT_{tx}$ | 0 | $-i(T_{tt} - T_{xx})$ | $-iT_{tx}$ |

Table 4: Action of the differential operators $\mathcal{D}_1, \mathcal{D}_2$ on the operators $K_\mu, T_{\mu\nu}$.

Since everything in Section 4.1 followed from the results of Table 1, and we recovered those results in this more general setting, we reach the same conclusions as in the rest of

that section. We conclude that we found additional evidence for the universality of the equations collected in Table 2.

To be explicit we summarize how to read off the results appropriate for the case at hand. Besides $T\bar{T}$, we can only consider the deformation by

$$
\begin{aligned}
\mathcal{O}_1 &\equiv 2\pi i K_{[t|}T_{\bar{z}|x]} \overset{\text{(free scalar)}}{=} J\bar{T} - \bar{J}\bar{\Theta}\,, \\
\mathcal{O}_2 &\equiv 2\pi i K_{[t|}T_{z|x]} \overset{\text{(free scalar)}}{=} \bar{J}T - J\Theta\,.
\end{aligned}
\tag{5.5}
$$

Then using also that $\bar{a} = a$, Table 2 collapses to Table 5. We obtained the latter table both from Table 2 using the rules explained and also by direct computation. Notably, only the bilinear composite operators make an appearance, and the linear operators are absent.

| $\mathcal{O}$ $\diagdown$ $+$ | $\mathcal{O}_1$ | $\mathcal{O}_2$ | $T\bar{T}$ |
|---|---|---|---|
| $\mathcal{O}_1$ | $1 - \frac{b}{2}$ | $-\frac{b}{2}$ | $0$ |
| $\mathcal{O}_2$ | $\frac{b}{2}$ | $1 + \frac{b}{2}$ | $0$ |
| $T\bar{T}$ | $-i\pi a$ | $-i\pi a$ | $1$ |

Table 5: The equation for $\frac{\partial}{\partial\lambda_{\mathcal{O}}}\mathcal{H}(\lambda, a, b)$ can be read off from the table in exactly the same way as from Table 2.

Continuing in this direction, we could obtain a flow equation for the spectrum in the same way as in Section 4.1. We do not write down the result of this straightforward exercise here. Unlike in the case of the deformed free scalar with shift symmetry, we do not have a point in the parameter space with a CFT with (anti)holomorphic currents, which was crucial in determining the initial conditions in Section 3, so we do not know how to determine the initial conditions for neither flow equations. This is the reason we only presented the treatment of the more general case as a check on the conjectured universality of the flow equations. The initial conditions could however be obtained in Gaussian theories: the massive complex boson and fermion, and it is an interesting future direction to obtain the spectrum of their irrelevant deformations.

## 5.2 A perturbative quantum check

The universal equations (4.18) and Table 2 for the Hamiltonian density $\mathcal{H} = -T_{tt}$ can be checked in quantum perturbation theory around a CFT, order by order in $\lambda$ and exactly in the background gauge fields $a$, $\bar{a}$ and $b$. These equations are statements about local operators modulo derivative terms, because they involve collision limits that are only defined up to derivatives.

In line with the rest of the paper we place the theory on $S^1 \times \mathbb{R}$ and work in the Hamiltonian formalism and on a fixed time slice.[4] We expand all local operators in Fourier modes. For example, the CFT's holomorphic stress-tensor is

$$
T_{\text{CFT}}(x) = -\left(\frac{2\pi}{L}\right)^2 \sum_{k=-\infty}^{\infty} e^{2\pi i k x/L}\,\ell_k\,, \qquad [\ell_k, \ell_m] = (k-m)\ell_{k+m} + \frac{c}{12}k^3\delta_{k+m,0}\,, \tag{5.6}
$$

---

[4]Translation to the path integral formalism should be straightforward.

520 in terms of shifted Virasoro modes $\ell_k \equiv L_k - \delta_{k,0} \, c/24$. See Appendix E for more conventions
521 and explicit formulas. All operators of interest are constructed from the dimensionless modes
522 $\ell_k$, $\bar{\ell}_k$, $j_k$, $\bar{j}_k$ of the CFT stress-tensor $T_{\mathrm{CFT}}(x)$, $\bar{T}_{\mathrm{CFT}}(x)$ and two independently-conserved
523 currents $J_{\mathrm{CFT}}(x)$, $\bar{J}_{\mathrm{CFT}}(x)$.

524    Schematically, one proceeds as follows. First turn on $\lambda$. In our formalism, $T_{xt}$, $J_t$,
525 $\bar{J}_t$ are fixed. Once the mode expansions of $J_\mu$, $\bar{J}_\mu$ and $T_{\mu\nu}$ are known up to order $\lambda^{p-1}$,
526 one computes the bilinear operator by which to deform, for example the collision limit
527 "$J\bar{T}$" $= 2\pi i J_{[t|}T_{\bar{z}|x]}$, to deduce $T_{tt}$ hence $H = -\int dx\, T_{tt}$ to order $\lambda^p$. Then conservation
528 gives $\partial_x T_{tx}$, $\partial_x T_{xx}$, $\partial_x J_x$, $\partial_x \bar{J}_x$ thus gives all modes of $T_{tx}$, $T_{xx}$, $J_x$, $\bar{J}_x$ except their zero
529 modes (since $\partial_x e^{inx}$ vanishes for $n = 0$). Locality fixes these zero modes up to ambiguities
530 explained in Section 3.3: shifts by multiples of the identity. Then $a$, $\bar{a}$, $b$ are turned on
531 using the same steps.

532    The rest of the section spells out details. We introduce useful deformations of the modes
533 $\ell_k$, $j_k$, $\bar{\ell}_k$, $\bar{j}_k$ in Section 5.2.1. Next, we tackle the two key difficulties: finding OPEs such
534 as $2\pi i J_{[t|}T_{\bar{z}|x]}$ in Section 5.2.2, and finding zero modes of $T_{tx}$, $T_{xx}$, $J_x$, $\bar{J}_x$ in Section 5.2.3.
535 Section 5.2.4 summarizes all the steps needed to do perturbation theory in our setting.

536    For the $J\bar{T}$ deformation we performed calculations specified by the procedure up to
537 order $\lambda^2$, with $\bar{a} = 0$ and exactly in $a$, $b$, and confirmed the universal equation. At this
538 order quantum effects could have spoiled the equation but some coefficients cancel. Let
539 us see why quantum effects arise at this order and not before. Our quantum calculations
540 reduce to classical calculations by replacing all commutators by Poisson brackets, replacing
541 all collision limits of operators by (coincident-point) products of functions, and setting
542 $c = 0$. The last requirement comes from comparing the equal-time commutators

$$[\bar{T}_{\mathrm{CFT}}(x), \bar{T}_{\mathrm{CFT}}(y)] = -2\pi i\left(\frac{c}{12}\delta'''(x-y) + 2\bar{T}_{\mathrm{CFT}}(y)\delta'(x-y) - \partial_y \bar{T}_{\mathrm{CFT}}(y)\delta(x-y)\right) \quad (5.7)$$

543 and $[T_{\mathrm{CFT}}(x), T_{\mathrm{CFT}}(y)]$ to their classical Poisson bracket analogues which have no $(c/12)\delta'''(x-$
544 $y)$ term. Quantum perturbation theory expresses $T_{\mu\nu}(x)$ and $J_\mu(x)$ as series in $\lambda$ of sums
545 of composite operators built from the CFT operators $T_{\mathrm{CFT}}(x)$, $J_{\mathrm{CFT}}(x)$, $\bar{T}_{\mathrm{CFT}}(x)$. Dimen-
546 sional analysis restricts the set of operators that can appear. We are interested in terms
547 multiplying $c$. Factors of $c$ appear in commutators (5.7) multiplied by the distribution
548 $\delta'''(x-y)$, which involves two additional derivatives and one fewer stress-tensor compared
549 to other terms. In expressions of $J_\mu$ and $T_{\mu\nu}$, operators multiplying $c$ thus involve two
550 derivatives. For $T_{tt}$, dimensional analysis only allows $\partial_x^2 J_{\mathrm{CFT}}$ at order $\lambda$, and at order $\lambda^2$,
551 only $\partial_x^2 \bar{T}_{\mathrm{CFT}}$, $J_{\mathrm{CFT}} \partial_x^2 J_{\mathrm{CFT}}$, $\partial_x J_{\mathrm{CFT}} \partial_x J_{\mathrm{CFT}}$ and $\partial_x^2 T_{\mathrm{CFT}}$ (actually, the last of these is for-
552 bidden because commutators do not produce it). Since the universal equation is defined
553 modulo derivatives, derivative terms $\partial_x^2 J_{\mathrm{CFT}}$ and $\partial_x^2 \bar{T}_{\mathrm{CFT}}$ cannot spoil it. However, the
554 terms $J_{\mathrm{CFT}} \partial_x^2 J_{\mathrm{CFT}}$ and $\partial_x J_{\mathrm{CFT}} \partial_x J_{\mathrm{CFT}}$ could arise with different ($b$-dependent) coefficients,
555 thus fail to give a derivative. These terms would then affect energy levels. The outcome of
556 our calculation is that the terms have equal coefficients so that they combine into a total
557 derivative

$$J_{\mathrm{CFT}} \partial_x^2 J_{\mathrm{CFT}} + \partial_x J_{\mathrm{CFT}} \partial_x J_{\mathrm{CFT}} = \frac{1}{2}\partial_x^2 J_{\mathrm{CFT}}^2. \quad (5.8)$$

558 The universal equation is thus confirmed, as are the energy levels.

559    Note that this check is not subsumed in the comparison of $J\bar{T}$-deformed energy levels
560 at $a = b = 0$ with earlier literature. Indeed, these previous results were worked out by
561 imposing holomorphy of $J_\mu$ (our definitions of $J_\mu$ differ slightly, as discussed in Appendix D)
562 which cannot be imposed once we turn on the backgrounds $a$ and $b$.

### 5.2.1 Spectrum-generating operators

We now return to a general deformation by $T_{\mu\nu}$, $J_\mu$, $\bar{J}_\mu$ and their antisymmetric bilinear combinations, and we introduce operators $\Lambda_k$, $\Upsilon_k$, $\overline{\Lambda}_k$, $\overline{\Upsilon}_k$ that play an important role when computing OPEs later.[5] For brevity we choose notations adapted to deformations by a single bilinear operator, with a single coupling $\lambda$, but it is easy to generalize. We call "eigenstate" or "state in the spectrum" a joint eigenstate of the various conserved charges: energy $H$, momentum $P$ and charges $Q$, $\overline{Q}$.

Tracking the $\lambda$-dependence of eigenstates is impractical because one must determine how each eigenstate $\ell_{k_1} \ldots j_{m_1} \ldots \bar{\ell}_{n_1} \ldots \bar{j}_{p_1} \ldots |\text{primary}\rangle$ in the CFT evolves. Instead we track relations between these states. More precisely we construct perturbatively a family of operators (see (E.4) for $O(\lambda)$ terms)

$$\Lambda_k = \ell_k + O(\lambda), \qquad \Upsilon_k = j_k + O(\lambda), \qquad \overline{\Lambda}_k = \bar{\ell}_k + O(\lambda), \qquad \overline{\Upsilon}_k = \bar{j}_k + O(\lambda) \quad (5.9)$$

that generate the spectrum in the sense that acting on an eigenstate gives another eigenstate. These operators can be defined abstractly as the result of "conjugating" the original modes $\ell_k$, $j_k$, $\bar{\ell}_k$, $\bar{j}_k$ by the deformation. For any eigenstate $|n\rangle_\lambda$ that is the image of some CFT state $|n\rangle$ under the deformation, $\Lambda_k |n\rangle_\lambda$ is defined as the image of $\ell_k |n\rangle$ under the deformation, and likewise $\Upsilon_k |n\rangle_\lambda \equiv (j_k |n\rangle)_\lambda$ and $\overline{\Lambda}_k |n\rangle_\lambda \equiv (\bar{\ell}_k |n\rangle)_\lambda$ and $\overline{\Upsilon}_k |n\rangle_\lambda \equiv (\bar{j}_k |n\rangle)_\lambda$ are images of $j_k |n\rangle$, $\bar{\ell}_k |n\rangle$, $\bar{j}_k |n\rangle$ under the deformation.[6]

This abstract definition does not help compute $\Lambda_k$, $\Upsilon_k$, $\overline{\Lambda}_k$, $\overline{\Upsilon}_k$ but leads to various properties.

- Given a state $|n\rangle = |h, q, \bar{h}, \bar{q}\rangle$ in the CFT with $\ell_0$, $j_0$, $\bar{\ell}_0$, $\bar{j}_0$ eigenvalues $h$, $q$, $\bar{h}$, $\bar{q}$ respectively, its image under the flow obeys $\Lambda_0 |n\rangle_\lambda = (\ell_0 |n\rangle)_\lambda = h|n\rangle_\lambda$ and so on. In that sense, $\Lambda_0 \pm \overline{\Lambda}_0$, $\Upsilon_0$, $\overline{\Upsilon}_0$ acting on $|n\rangle_\lambda$ measure the energy, momentum and charges of the original state $|n\rangle$.

- Since charge and momentum of states are fixed, $\Upsilon_0 = j_0$, $\overline{\Upsilon}_0 = \bar{j}_0$, and $\Lambda_0 - \overline{\Lambda}_0 = \ell_0 - \bar{\ell}_0$.

- The operators obey the same Virasoro and Kač–Moody algebra as $\ell_k$, $j_k$, $\bar{\ell}_k$, $\bar{j}_k$, namely $[\Lambda_k, \overline{\Lambda}_m] = [\Lambda_k, \overline{\Upsilon}_m] = [\Upsilon_k, \overline{\Lambda}_m] = [\Upsilon_k, \overline{\Upsilon}_m] = 0$ and

$$[\Lambda_k, \Lambda_m] = (k - m)\Lambda_{k+m} + \frac{c}{12}k^3\delta_{k+m,0}, \quad [\Lambda_k, \Upsilon_m] = -m\Upsilon_{k+m}, \quad [\Upsilon_k, \Upsilon_m] = k\delta_{k+m},$$

$$[\overline{\Lambda}_k, \overline{\Lambda}_m] = (k - m)\overline{\Lambda}_{k+m} + \frac{c}{12}k^3\delta_{k+m,0}, \quad [\overline{\Lambda}_k, \overline{\Upsilon}_m] = -m\overline{\Upsilon}_{k+m}, \quad [\overline{\Upsilon}_k, \overline{\Upsilon}_m] = k\delta_{k+m}.$$
$$(5.10)$$

- Acting with $\Lambda_k$ or $\Upsilon_k$ or $\overline{\Lambda}_k$ or $\overline{\Upsilon}_k$ on an eigenstate $|n\rangle_\lambda$ gives another eigenstate. Its energy is higher than that of $|n\rangle_\lambda$ if $k < 0$ and lower if $k > 0$. One could call these operators "raising" or "lowering" operators according to the sign of $k$, but importantly their existence does not make the spectrum trivial. Indeed, energies of different eigenstates are shifted by different amounts.

Explicit low-order perturbative calculations suggest a last property for our class of deformations.

---

[5]In the case of the $J\bar{T}$ deformation, the operators $\overline{\Lambda}_k$ should reduce to effectively non-local state-dependent Virasoro generators found previously in [24, 27].

[6]More precisely, the CFT spectrum has states with degenerate energy and momentum and charge, for instance $\ell_{-4}|0\rangle$ and $\ell_{-2}^2|0\rangle$, and to distinguish $(\ell_{-4}|0\rangle)_\lambda$ from $(\ell_{-2}^2|0\rangle)_\lambda$ one uses KdV conserved charges, under which the CFT spectrum is non-degenerate. These KdV conserved charges also exist in the deformed theory for any deformation in the class we consider, which makes $\Lambda_k$, $\Upsilon_k$, $\overline{\Lambda}_k$, $\overline{\Upsilon}_k$ well-defined.

- The Hamiltonian $H$ can be written as a function $\mathsf{H}(\lambda; \Lambda_0, \Upsilon_0, \overline{\Lambda}_0, \overline{\Upsilon}_0) = \frac{2\pi}{L}(\Lambda_0 + \overline{\Lambda}_0) + O(\lambda)$, given explicitly for the $J\bar{T}$ deformation in (E.5). In particular, eigenstates of $H$, $P$, $Q$, $\overline{Q}$ are the same as eigenstates of $\Lambda_0$, $\Upsilon_0$, $\overline{\Lambda}_0$, $\overline{\Upsilon}_0$. From it we deduce that the energy of a state $|h, q, \bar{h}, \bar{q}\rangle_\lambda$ is $\mathsf{H}(\lambda; h, q, \bar{h}, \bar{q})$ since

$$H|h, q, \bar{h}, \bar{q}\rangle_\lambda = \mathsf{H}(\lambda; \Lambda_0, \Upsilon_0, \overline{\Lambda}_0, \overline{\Upsilon}_0)|h, q, \bar{h}, \bar{q}\rangle_\lambda = \mathsf{H}(\lambda; h, q, \bar{h}, \bar{q})|h, q, \bar{h}, \bar{q}\rangle_\lambda. \quad (5.11)$$

Energy levels then depend on the original energy, momentum and charges in the same way as the Hamiltonian depends on $\Lambda_0 \pm \overline{\Lambda}_0$, $\Upsilon_0$, $\overline{\Upsilon}_0$. Reversing the logic, our solution (6.4) for energy levels thus predicts the exact Hamiltonian. For example for the $J\bar{T}$-deformed CFT with $a = \bar{a} = b = 0$, we expect

$$H \overset{\text{prediction}}{=} \frac{2\pi}{L}\left(\Lambda_0 - \overline{\Lambda}_0 - \frac{L^2}{2\pi^4\lambda^2}\left(1 - \frac{2\pi^2 i\lambda}{L}\Upsilon_0 - \sqrt{\left(1 - 2\pi^2 i(\lambda/L)\Upsilon_0\right)^2 - 2\left(2\pi^2 i\lambda/L\right)^2\overline{\Lambda}_0}\right)\right). \quad (5.12)$$

How do we find the expressions of the spectrum-generating operators $\Lambda_k$, $\Upsilon_k$, $\overline{\Lambda}_k$, $\overline{\Upsilon}_k$ order-by-order in $\lambda$ in terms of the CFT modes $\ell_k$, $j_k$, $\bar{\ell}_k$, $\bar{j}_k$? The construction of $\partial_\lambda\Lambda_k$, $\partial_\lambda\Upsilon_k$, $\partial_\lambda\overline{\Lambda}_k$, $\partial_\lambda\overline{\Upsilon}_k$ is easiest to do in terms of the spectrum-generating operators themselves; it can then be translated to the CFT modes using expressions of $\Lambda_k$, $\Upsilon_k$, $\overline{\Lambda}_k$, $\overline{\Upsilon}_k$ at the previous order in $\lambda$.

While in practice we eventually do all of our calculations in terms of $\Lambda_k$, $\Upsilon_k$, $\overline{\Lambda}_k$, $\overline{\Upsilon}_k$, derivatives with respect to couplings always denote derivatives at fixed $\ell_k$, $j_k$, $\bar{\ell}_k$, $\bar{j}_k$. This makes it a bit awkward to reconstruct an operator $\mathcal{O} = \sum_{n\geq 0} \frac{1}{n!}\lambda^n \mathcal{O}^{(n)}$ from its $\lambda$ derivative because the $\lambda^p/p!$ term in $\partial_\lambda\mathcal{O}$ works out to be

$$(\partial_\lambda\mathcal{O})^{(p)} = \mathcal{O}^{(p+1)} + \sum_{n=0}^{p}\binom{p}{n}\left(\partial_\lambda(\mathcal{O}^{(n)})\right)^{(p-n)}. \quad (5.13)$$

Note that we had to expand $\partial_\lambda(\mathcal{O}^{(n)})$ in powers of $\lambda$ because it involves derivatives of $\Lambda_k$, $\Upsilon_k$, $\overline{\Lambda}_k$, $\overline{\Upsilon}_k$ that are themselves series in $\lambda$.

To proceed, we first note that $\partial_\lambda\Upsilon_0 = \partial_\lambda\overline{\Upsilon}_0 = \partial_\lambda(\Lambda_0 - \overline{\Lambda}_0) = 0$ by charge and momentum conservation. Then we construct $\partial_\lambda\overline{\Lambda}_0 = \partial_\lambda\Lambda_0$ such that (5.12) holds (or its analogue for other deformations). We find it by solving (5.12) for $\overline{\Lambda}_0$ in terms of $H$ and $\Upsilon_0$ and $\Lambda_0 - \overline{\Lambda}_0$,

$$\overline{\Lambda}_0 = \frac{1}{2}\left(1 - \frac{2\pi^2 i\lambda}{L}\Upsilon_0\right)\left(\frac{LH}{2\pi} - \Lambda_0 + \overline{\Lambda}_0\right) + \frac{\pi^4\lambda^2}{2L^2}\left(\frac{LH}{2\pi} - \Lambda_0 + \overline{\Lambda}_0\right)^2, \quad (5.14)$$

and taking a $\partial_\lambda$ derivative. The deformation $\partial_\lambda H$ commutes with $\Lambda_0 - \overline{\Lambda}_0 = \ell_0 - \bar{\ell}_0$ (is translation-invariant) so $\partial_\lambda\overline{\Lambda}_0$ also does, namely all terms $\Lambda_{m_1}\ldots\Upsilon_{n_1}\ldots\overline{\Lambda}_{\bar{m}_1}\ldots\overline{\Upsilon}_{\bar{n}_1}\ldots$ in $\partial_\lambda\overline{\Lambda}_0$ obey $\sum m + \sum n = \sum \bar{m} + \sum \bar{n}$. At the orders we checked we additionally find that there are no terms that commute with $\Lambda_0$ (or equivalently with $\overline{\Lambda}_0$), namely no term with

$$\sum m + \sum n = \sum \bar{m} + \sum \bar{n} = 0. \quad (5.15)$$

The lack of such terms is essential for the following construction to work. We now know $\partial_\lambda\Lambda_0 = \partial_\lambda\overline{\Lambda}_0$ up to a certain order in $\lambda$ and want to construct other $\partial_\lambda\Lambda_k$, $\partial_\lambda\Upsilon_k$, $\partial_\lambda\overline{\Lambda}_k$, $\partial_\lambda\overline{\Upsilon}_k$ that are consistent with the commutators (5.10).

First, we want to preserve $[\overline{\Lambda}_0, \Lambda_k] = [\overline{\Lambda}_0, \Upsilon_k] = 0$. From their derivatives we learn that we need

$$[\overline{\Lambda}_0, \partial_\lambda\Lambda_k] = [\Lambda_k, \partial_\lambda\overline{\Lambda}_0], \quad \text{and} \quad [\overline{\Lambda}_0, \partial_\lambda\Upsilon_k] = [\Upsilon_k, \partial_\lambda\overline{\Lambda}_0]. \quad (5.16)$$

Crucially, the right-hand sides do not contain any term of the form $\Lambda\ldots\Upsilon\ldots\overline{\Lambda}_{m_1}\ldots\overline{\Upsilon}_{n_1}\ldots$ with $\sum \bar{m} + \sum \bar{n} = 0$, because as we mentioned, $\partial_\lambda\overline{\Lambda}_0$ do not contain such terms. Then (5.16)

631 fixes $\partial_\lambda \Lambda_k$ and $\partial_\lambda \Upsilon_k$ up to such terms, and we choose to define $\partial_\lambda \Lambda_k$ and $\partial_\lambda \Upsilon_k$ without
632 any such term, even though we could add arbitrary such terms without spoiling (5.16).

633   We define $\partial_\lambda \overline{\Lambda}_k$ and $\partial_\lambda \overline{\Upsilon}_k$ similarly, based on $[\overline{\Lambda}_0, \overline{\Lambda}_k] = -k\overline{\Lambda}_k$, which gives $[\overline{\Lambda}_0, \partial_\lambda \overline{\Lambda}_k] +$
634 $k \partial_\lambda \overline{\Lambda}_k = [\overline{\Lambda}_k, \partial_\lambda \overline{\Lambda}_0]$ hence fixes $\partial_\lambda \overline{\Lambda}_k$ up to terms of the form $\Lambda_{m_1} \ldots \Upsilon_{n_1} \ldots \overline{\Lambda}_{\bar{m}_1} \ldots \overline{\Upsilon}_{\bar{n}_1} \ldots$
635 with $\sum \bar{m} + \sum \bar{n} = k$. We choose to define $\partial_\lambda \overline{\Lambda}_k$ without any such term. Equivalently, these
636 terms are characterized by $\sum m + \sum n = 0$, so this is really the analogue of the condition
637 we put on terms appearing in $\partial_\lambda \Lambda_k$ and $\partial_\lambda \Upsilon_k$.

638   These definitions reduce for $k = 0$ to the ones we already imposed.

639   Finally we must check our constructed $\partial_\lambda \Lambda_k$, $\partial_\lambda \Upsilon_k$, $\partial_\lambda \overline{\Lambda}_k$, $\partial_\lambda \overline{\Upsilon}_k$ give rise to the remaining
640 commutators (5.10). Let us just show one calculation explicitly: that $\partial_\lambda \big([\Lambda_k, \Lambda_m] - (k -$
641 $m)\Lambda_{k+m} - k^3 \delta_{k+m,0}\, c/12\big)$ vanishes. First, note that this derivative is built from some
642 $\partial_\lambda \Lambda_n$, which by construction have no terms that commute with $\overline{\Lambda}_0$, so it is enough to
643 check that $[\overline{\Lambda}_0, \partial_\lambda(\ldots)]$ vanishes. We compute (at an order in $\lambda$ at which we know the
644 commutators (5.10) but not yet their $\lambda$ derivative)

$$
\begin{aligned}
[\overline{\Lambda}_0, \partial_\lambda(\ldots)] &= [\overline{\Lambda}_0, [\partial_\lambda \Lambda_k, \Lambda_m]] + [\overline{\Lambda}_0, [\Lambda_k, \partial_\lambda \Lambda_m]] - (k - m)[\overline{\Lambda}_0, \partial_\lambda \Lambda_{k+m}] \\
&= [[\overline{\Lambda}_0, \partial_\lambda \Lambda_k], \Lambda_m] + [\Lambda_k, [\overline{\Lambda}_0, \partial_\lambda \Lambda_m]] - (k - m)[\overline{\Lambda}_0, \partial_\lambda \Lambda_{k+m}] \\
&= [[\Lambda_k, \partial_\lambda \overline{\Lambda}_0], \Lambda_m] + [\Lambda_k, [\Lambda_m, \partial_\lambda \overline{\Lambda}_0]] - (k - m)[\Lambda_{k+m}, \partial_\lambda \overline{\Lambda}_0] \\
&= \big[[\Lambda_k, \Lambda_m] - (k - m)\Lambda_{k+m}, \partial_\lambda \overline{\Lambda}_0\big] = 0.
\end{aligned}
\tag{5.17}
$$

645   This concludes the construction of spectrum-generating operators $\Lambda_k$, $\Upsilon_k$, $\overline{\Lambda}_k$, $\overline{\Upsilon}_k$. At
646 each order in $\lambda$ one should check that $\partial_\lambda \overline{\Lambda}_0$ deduced from (5.14) has no term $\Lambda \ldots \Upsilon \ldots \overline{\Lambda} \ldots \overline{\Upsilon} \ldots$
647 that commutes with $\Lambda_0$. Other properties of these operators then come for free.

648 ## 5.2.2   Computing OPEs

649 The Virasoro (and Kač–Moody) algebras (5.6) and (E.3) obeyed by $\ell_k$, $j_k$, $\bar{\ell}_k$, $\bar{j}_k$ are
650 unchanged by the deformation, and the same is true for commutators of local operators
651 such as $T_{\text{CFT}}(x)$ whose expression in terms of modes does not depend on couplings.

652   On the other hand, OPEs of such coupling-independent operators change.[7] For instance,
653

$$
J_{\text{CFT}}(x)\bar{T}_{\text{CFT}}(y) = 2\pi\lambda\left(\frac{c/2}{(x-y)^4} + \frac{2\bar{T}_{\text{CFT}}(y)}{(x-y)^2} + \frac{\partial_y \bar{T}_{\text{CFT}}(y)}{(x-y)}\right) + O((x-y)^0) + O(\lambda^2) \tag{5.18}
$$

654 in the $J\bar{T}$-deformed theory, even though the left-hand side has no $\lambda$ dependence whatsoever.
655 In our formalism, this seemingly contradictory result comes from how the notion of well-
656 defined operator depends on $\lambda$. In the CFT,

$$
J_{\text{CFT}}(x)\bar{T}_{\text{CFT}}(x) = i\left(\frac{2\pi}{L}\right)^3 \sum_{k,m} e^{2\pi i(k+m)x/L} j_k \bar{\ell}_m \tag{5.19}
$$

657 is well-defined, in the sense that each mode ($k + m = $ constant) is an infinite sum that
658 truncates when acting on any state in the spectrum.[8] The spectrum depends on $\lambda$ and
659 in the deformed theory the sum fails to truncate, so that the coincident-point operator
660 $J_{\text{CFT}}(x)\bar{T}_{\text{CFT}}(x)$ is ill-defined. The correct OPE (5.18) can be checked in principle by

---

[7]This is a rather different situation than the OPEs considered in [23], because what these authors denote $T, \Theta, \bar{\Theta}, \bar{T}$ are certain components of the deformed stress-tensor $T_{\mu\nu}$, whereas here we consider OPEs, in the deformed theory, of the CFT operators.

[8]By "state in the spectrum" we mean an eigenstate of the Hamiltonian, momentum, and conserved charge.

661  comparing matrix elements $_\lambda\langle n|J_{\mathrm{CFT}}(x)\bar{T}_{\mathrm{CFT}}(y)|n'\rangle_\lambda$ between eigenstates $|n\rangle_\lambda$, $|n'\rangle_\lambda$ to
662  matrix elements of the right-hand side.

663      Let us briefly discuss collision limits in a CFT when working explicitly in modes. First
664  consider $J_{\mathrm{CFT}}(x)J_{\mathrm{CFT}}(y)$. To get a finite collision limit one reorders modes using the
665  commutator (we set $L = 2\pi$ to shorten expressions)

$$J_{\mathrm{CFT}}(x)J_{\mathrm{CFT}}(y) = -\sum_{k,m}e^{i(kx+my)}j_k j_m = -\sum_{k>m}e^{i(kx+my)}[j_k,j_m] - \sum_{k,m}e^{i(kx+my)}\,{:}j_k j_m{:}$$

$$= \frac{1}{(2\sin\frac{x-y}{2})^2} - \sum_{k,m}e^{i(kx+my)}\,{:}j_k j_m{:} = \frac{1}{(x-y)^2} + \frac{1}{12} + {:}J_{\mathrm{CFT}}(y)J_{\mathrm{CFT}}(y){:} + O(x-y)$$

(5.20)

666  where ${:}j_k j_m{:} = (j_k j_m \text{ if } k < m \text{ else } j_m j_k)$.[9] The reason $-\sum_{k,m}e^{i(k+m)x}\,{:}j_k j_m{:}$ has finite
667  matrix elements in any energy eigenstate $|n\rangle$ of the CFT is that ${:}j_k j_m{:}|n\rangle$ vanishes for large
668  enough $k$ or $m$ (thanks to the normal-ordering) while $\langle n'|\,{:}j_k j_m{:}$ vanishes for negative enough
669  $k$ or $m$. Altogether only finitely many $k$ and $m$ can contribute to a given $\langle n'|{:}j_k j_m{:}|n\rangle$.
670  For more complicated examples such as collisions of Sugawara stress-tensors $\frac{1}{2}\,{:}J_{\mathrm{CFT}}^2{:}$, the
671  prescription is still to reorder modes using commutators until modes are all ordered, then
672  evaluate the series such as $\sum_k e^{ik(x-y)}k$ that arise. Normal-ordered products have finite
673  collision limits. In a CFT on the plane, the shortcut to get the regularized collision limits
674  of operators such as ${:}J_{\mathrm{CFT}}^p{:}$ is simply to normal-order the modes and take $x = y$.

675      Consider now the collision of a product $\mathcal{A}(x)\mathcal{B}(y)$ of local operators[10] in the deformed
676  theory.

677      We can apply a similar idea: express $\mathcal{A}$ and $\mathcal{B}$ in terms of spectrum-generating operators
678  $\Lambda_k$, $\Upsilon_k$, $\overline{\Lambda}_k$, $\overline{\Upsilon}_k$ then sort these operators by increasing $k$. Let us call the resulting normal-
679  ordered product $\mathrm{Sort}(\mathcal{A}(x)\mathcal{B}(y))$. Since this places lowering operators to the right of raising
680  ones, all matrix elements in energy eigenstates truncate the sums to finitely many terms,
681  hence remain finite as $x \to y$. The $x \to y$ collision limit $\mathrm{Sort}(\mathcal{A}\mathcal{B})$ is thus well-defined.
682  Unfortunately, this ordering prescription is not consistent with locality, namely we find by
683  explicit calculations that the commutator of $\mathrm{Sort}(\mathcal{A}\mathcal{B})(y)$ with a local operator at $w$ fails
684  to vanish for $w \neq y$.

685      To preserve locality we cannot use the shortcut of normal ordering. Instead, we
686  keep track of all commutators when reordering the operators $\Lambda_k$, $\Upsilon_k$, $\overline{\Lambda}_k$, $\overline{\Upsilon}_k$ as we did
687  in (5.20) in the CFT case. Once all terms are ordered, the coefficient of each product
688  $\Upsilon\ldots\Lambda\ldots\overline{\Lambda}\ldots\overline{\Upsilon}\ldots$, often an infinite sum, should be evaluated and expanded as $x \to y$.
689  The sought-after collision limit is then the $(x-y)^0$ term. Besides the normal-ordered
690  product $\mathrm{Sort}(\mathcal{A}\mathcal{B})$ it may include additional terms similar to the shift by $1/12$ in (5.20).

691      We computed the non-trivial OPE (5.18) of the CFT local operators $J_{\mathrm{CFT}}(x)$ and $\bar{T}_{\mathrm{CFT}}(y)$
692  by following these steps in the $J\bar{T}$-deformed theory. Converting from modes $j_k$ and $\bar{\ell}_k$
693  to operators $\Lambda_k$, $\Upsilon_k$, $\overline{\Lambda}_k$ uses (the inverse of) the explicit formulas (E.4). At order $\lambda$,
694  $J_{\mathrm{CFT}}(x)\bar{T}_{\mathrm{CFT}}(y)$ includes terms such as $\sum_{k,m,n}(\ldots)\Upsilon_k\Upsilon_m\overline{\Lambda}_n$ in which the $\Upsilon$ must be
695  reordered. The commutator terms $[\Upsilon_k,\Upsilon_m]\overline{\Lambda}_n$ give sums of modes $\overline{\Lambda}_n$ whose coefficients
696  are singular as $x \to y$, which lead to $\bar{T}_{\mathrm{CFT}}$ and $\partial_y\bar{T}_{\mathrm{CFT}}$ terms in (5.18). The $c$-dependence
697  in the OPE comes directly from the $c$-dependence of the dictionary (E.4) between CFT
698  modes and deformed ones $\Lambda_k$, $\Upsilon_k$, $\overline{\Lambda}_k$.

---

[9]The shift by $1/12$ is the expected shift $\ell_n = L_n - (c/24)\delta_{n,0}$ once one remembers that ${:}J_{\mathrm{CFT}}^2{:}$ is twice
the Sugawara stress-tensor, which has central charge $c = 1$ in this case.

[10]The deformed operators $J_\mu(x)$, $\bar{J}_\mu(x)$, $T_{\mu\nu}(x)$ are eventually built from various collision limits at $x$ of
the CFT operators $J_{\mathrm{CFT}}$, $\bar{J}_{\mathrm{CFT}}$, $T_{\mathrm{CFT}}$, $\bar{T}_{\mathrm{CFT}}$ and their derivatives, so commutators of two such operators
at different points $x_1$ and $x_2$ vanish, namely these operators are still local after deformations. The fact that
the $T\bar{T}$ deformation preserves locality was already observed in [23].

699    The OPE of $J_{\text{CFT}}(x)\bar{T}_{\text{CFT}}(y)$ is only one term in the OPE $2\pi i J_{[t|}T_{\bar{z}|x]}$ that we are
700 really interested in, because the components $J_\mu$ and $T_{\mu\nu}$ depend on $\lambda$. Among other terms,
701 $J_x$ contains $\lambda\bar{T}_{\text{CFT}}$, whose OPE with $\bar{T}_{\text{CFT}}$ cancels most terms in (E.4). Altogether, the
702 collision limit we care about works out to be

$$2\pi i J_{[t|}(x)T_{\bar{z}|x]}(y) = -\lambda\frac{\pi\partial_y\bar{T}_{\text{CFT}}(y)}{x-y} + O((x-y)^0) + O(\lambda^2). \qquad (5.21)$$

703 The operator $\partial_y\bar{T}_{\text{CFT}}(y)$ is a derivative, as expected from general considerations about
704 antisymmetric combinations of conserved currents. We are actually interested in the $(x-y)^0$
705 term in this OPE. Working it out we got a finite collision limit expressed in terms of $\Lambda_k$,
706 $\Upsilon_k$, $\bar{\Lambda}_k$.
707    By definition, $\partial_\lambda T_{tt} = -2\pi i J_{[t|}T_{\bar{z}|x]}$, where the $\lambda$ derivative is taken at fixed $\ell_k$, $j_k$, $\bar{\ell}_k$.
708 This lets us get the next power of $\lambda$ in $T_{tt}$, by either translating $2\pi i J_{[t|}T_{\bar{z}|x]}$ to the modes
709 $\ell_k$, $j_k$, $\bar{\ell}_k$, or accounting for non-zero $\partial_\lambda\Lambda_k$, $\partial_\lambda\Upsilon_k$, $\partial_\lambda\bar{\Lambda}_k$.

### 5.2.3   Using background fields to get local currents

711 Once $T_{tt}$ is known, conservation equations give $\partial_x J_x$, $\partial_x\bar{J}_x$, $\partial_x T_{tx}$, $\partial_x T_{xx}$, but give no
712 information on zero modes of these spatial components of currents. Finding the zero modes
713 is absolutely crucial because they affect all modes of bilinear products such as $2\pi i J_{[t|}T_{\bar{z}|x]}$,
714 used to define $T_{tt}$ at the next order in $\lambda$. In principle one should impose locality to find
715 these modes, namely one should ask for the commutator with CFT operators $T_{\text{CFT}}$, $J_{\text{CFT}}$,
716 $\bar{T}_{\text{CFT}}$, $\bar{J}_{\text{CFT}}$ to be zero at separated points. This is very difficult: if we work in terms of $\Lambda_k$,
717 $\Upsilon_k$, $\bar{\Lambda}_k$, $\bar{\Upsilon}_k$ then commutators with modes of $T_{\text{CFT}}$, $J_{\text{CFT}}$, $\bar{T}_{\text{CFT}}$, $\bar{J}_{\text{CFT}}$ are complicated;
718 and if we work in terms of $\ell_k$, $j_k$, $\bar{\ell}_k$, $\bar{j}_k$ there is no good way to determine whether a given
719 sum of products of modes is well-defined, as we discussed near (5.19).
720    To get around this hurdle, and to turn on $a$ and $b$, we treat our classical evolution
721 equation (4.18) or Table 2 as providing an Ansatz for $T_{tt}(\lambda, a, \bar{a}, b)$ hence for $J_x = i\partial_a T_{tt}$ and
722 $\bar{J}_x = -i\partial_{\bar{a}}T_{tt}$ and $T_{tx} = -i\partial_b T_{tt}$ and $T_{xx} = \partial_L(LT_{tt})$. As everything else in this subsection,
723 checking the Ansatz is done order by order in $\lambda$, so let us assume that $T_{tt}(\lambda, a, \bar{a}, b)$ is
724 known up to order $\lambda^{p-1}$, and exactly in $a$, $\bar{a}$, $b$.
725    The order $\lambda^p$ term of $T_{tt}$ provided by our classical equation is correct for $a = \bar{a} = b = 0$
726 by definition of the deformation. Then, to show that $T_{tt}(\lambda, a, \bar{a}, b)$ matches the definition
727 of the $a$, $\bar{a}$, $b$ deformations, we need only check that for any $(a, \bar{a}, b)$ the derivatives $i\partial_a T_{tt}$,
728 $-i\partial_{\bar{a}}T_{tt}$ and $-i\partial_b T_{tt}$ are indeed equal to the correct components $J_x$, $\bar{J}_x$, $T_{tx}$. These are
729 characterized (up to shifts by multiples of the identity discussed in Section 3.3) by the
730 conservation equations and by locality. Locality is automatic because $T_{tt}$ is constructed
731 from local operators (including collisions, computed as explained above), and taking $a$, $\bar{a}$, $b$
732 derivatives commutes with taking a commutator with the reference local (CFT) operators
733 $T_{\text{CFT}}$, $J_{\text{CFT}}$, $\bar{T}_{\text{CFT}}$, $\bar{J}_{\text{CFT}}$. On the other hand, we do not have a general proof of conservation,
734 so one has to check at each order in $\lambda$ that the Ansatz obeys conservation, using explicit
735 expressions for a given deformation.

### 5.2.4   Summary of the procedure

737 To start the whole process we need to know the "initial data": $J_\mu$, $\bar{J}_\mu$, $T_{\mu\nu}$ at order
738 $\lambda^0$ for all $a$, $\bar{a}$, $b$ (and $L$). At this order, the stress-tensor and conserved current are
739 linear combinations (3.7) of the CFT ones. The dependence on $a$, $\bar{a}$, $b$ is fixed up to the
740 ambiguities (3.11) under shifts by multiples of the identity. We also keep $\Lambda_k = \ell_k + O(\lambda)$
741 etc. with no $a$ nor $b$ dependence at order $\lambda^0$.

One safe way to avoid accidentally writing ill-defined products such as (5.19) is to work in terms of the spectrum-generating operators $\Lambda_k$, $\Upsilon_k$, $\overline{\Lambda}_k$, $\overline{\Upsilon}_k$ and systematically commute operators with larger $k$ towards the right of any product. We use (5.13) in the form $\mathcal{O}^{(p)} = (\partial_\lambda \mathcal{O})^{(p-1)} - \cdots$ to deduce an operator at order $\lambda^p$ from its derivative at order $\lambda^{p-1}$.

The concrete procedure to get the order $\lambda^p$ terms in $J_\mu$, $\bar{J}_\mu$, $T_{\mu\nu}$ (hence in $H$) knowing their order $\lambda^{p-1}$ terms is then as follows.

1. Determine up to order $\lambda^{p-1}$ the collision limit $2\pi i J_{[t|} T_{\bar{z}|x]} + \pi b J_{[t|} T_{t|x]} + \ldots$ appearing on the right-hand side of (4.18) or its generalizations from Table 2. For $a = \bar{a} = b = 0$ this is $\partial_\lambda T_{tt}$, while for nonzero $(a, \bar{a}, b)$ it is only an Ansatz, checked later. To deal with derivative ambiguities, one includes with unknown coefficients the derivative of every local operator allowed by dimensional analysis.

2. Write the expression for $\partial_\lambda \overline{\Lambda}_0$ given in (5.14) or its generalizations including background fields.
   **Check** that this Ansatz is valid, in that it produces no terms that commute with $\Lambda_0$ (or equivalently with $\overline{\Lambda}_0$). Deduce all other $\partial_\lambda \Lambda_k$, $\partial_\lambda \Upsilon_k$, $\partial_\lambda \overline{\Lambda}_k$, $\partial_\lambda \overline{\Upsilon}_k$ up to order $\lambda^{p-1}$.

3. Use (5.13) to deduce $T_{tt}^{(p)}$ from $(\partial_\lambda T_{tt})^{(p-1)}$ computed in step 1 and from the order $\lambda^{p-1-k}$ terms of derivatives $\partial_\lambda(T_{tt}^{(k)})$, $0 \le k \le p-1$, which involve terms computed in step 2. Deduce $H^{(p)}$. By construction of $\partial_\lambda \overline{\Lambda}_0$ the Hamiltonian is expressed in terms of $\Lambda_0$, $\Upsilon_0$, $\overline{\Lambda}_0$, $\overline{\Upsilon}_0$.

4. Likewise, work out $\partial_a$, $\partial_{\bar{a}}$, $\partial_b$ derivatives of $\Lambda_k$, $\Upsilon_k$, $\overline{\Lambda}_k$, $\overline{\Upsilon}_k$ up to order $\lambda^p$ by noticing that their $\partial_\lambda$ derivative is known to order $\lambda^{p-1}$. For instance $\partial_\lambda \partial_a \Lambda_k = \partial_a \partial_\lambda \Lambda_k = \partial_a(\text{known} + O(\lambda^p))$, which only requires knowing $\partial_a$ derivatives of $\Lambda_k$, $\Upsilon_k$, $\overline{\Lambda}_k$, $\overline{\Upsilon}_k$ up to order $\lambda^{p-1}$.

5. Compute $J_x = i\partial_a T_{tt}$ and $\bar{J}_x = -i\partial_{\bar{a}} T_{tt}$ and $T_{tx} = -i\partial_b T_{tt}$ and $T_{xx} = \partial_L(L T_{tt})$ up to $\lambda^{p-1}$. **Check** current conservation $[H, J_t] + \partial_x J_x = 0$ and similarly for $\bar{J}_\nu$ and $T_{\mu\nu}$. This check fixes the unknown derivative terms from step 1. As explained in Section 5.2.3, locality is automatic and the check proves that the Ansatz for $T_{tt}^{(p)}$ is correct.

We performed this procedure (without $\bar{J}_{\text{CFT}}$) up to $p = 2$ for the $J\bar{T}$ deformation, thus checking the universal equation (4.18) as a local operator equation modulo derivatives. This proves to order $\lambda^2$ that the spectrum, including its dependence on background gauge fields, is exactly as predicted by the evolution equation (4.21), which we solve exactly in (6.1) and (6.4).

The procedure is quite bulky, and needs to be simplified before it can be pushed to much higher order. Perhaps the path integral formalism can help, but one would have to carefully work out OPEs such as (5.21) in this approach. In addition, the spectrum-generating operators $\Lambda_k$, $\Upsilon_k$, $\overline{\Lambda}_k$, $\overline{\Upsilon}_k$ seem to be less natural in the path integral than in the Hamiltonian formalism.

In this discussion we worked with spectrum-generating operators $\Lambda_k$, $\Upsilon_k$, $\overline{\Lambda}_k$, $\overline{\Upsilon}_k$ to make writing normal ordered products easier, but we nevertheless considered $\ell_k$, $j_k$, $\bar{\ell}_k$, $\bar{j}_k$ as fixed when defining derivatives such as $\partial_\lambda$. Just as one switches from the Schrödinger to the Heisenberg picture in quantum mechanics, we could switch from having $\lambda$-dependent states $|n\rangle_\lambda$, hence $\lambda$-dependent spectrum-generating operators, to having a $\lambda$-independent spectrum and spectrum-generating operators. This would mean $\partial_\lambda \Lambda_k$ and so on would vanish, while $\partial_\lambda \ell_k$ etc. would not vanish any longer. It may be instructive to translate our universal equation to this picture.

## 6  The spectrum from the solution of flow equations

### 6.1  Solving a large family of theories

While (4.23) is a rather intimidating nonlinear PDE of five variables, we will nevertheless write down a closed form solution of it. While we suspect that there should be a straightforward derivation of the solution, we first obtained the solution below using intuition from known results and solving the equations in series form. Before presenting the solution, we sketch the steps that led us to it.

In our conventions (with $g_{J\bar{T}} = 1$, namely $\lambda_{J\bar{T}} = i\ell$, see (4.22)), the spectrum of the $J\bar{T}$-deformed CFT in [5] is

$$
E_n L = \frac{1}{\pi^3 \left(\frac{\ell}{L}\right)^2} \left( 1 + \pi Q \left(\frac{\ell}{L}\right) + \pi^3 p \left(\frac{\ell}{L}\right)^2 - \sqrt{1 + 2\pi Q \left(\frac{\ell}{L}\right) - (2\pi^3(\epsilon_0 - p) - \pi^2 Q^2) \left(\frac{\ell}{L}\right)^2} \right),
$$
$$
p \equiv P_n L, \qquad \epsilon_0 \equiv E_n^{(0)} L.
$$
(6.1)

The spectrum of the $T\bar{T}$-deformed CFT is (with $g_{T\bar{T}} = 1$ or $\lambda_{T\bar{T}} = \ell^2$) [2,3]:

$$
E_n L = \frac{1}{2\pi^2 \left(\frac{\ell}{L}\right)^2} \left( 1 - \sqrt{1 - 4\pi^2 \epsilon_0 \left(\frac{\ell}{L}\right)^2 + 4\pi^4 p^2 \left(\frac{\ell}{L}\right)^4} \right).
$$
(6.2)

The spectrum can be checked to obey the Burgers equation (2.5), if we use the relation $\lambda_{(2.5)} = -2\pi^2 \ell^2$ as explained around (2.1). Based on these examples, a reasonable guess is that the spectrum of the full theory with all couplings turned on may take the form:

$$
E_n L = \frac{1}{\# \left(\frac{\ell}{L}\right)^2} \left( P_2 \left(\frac{\ell}{L}\right) - \sqrt{P_4 \left(\frac{\ell}{L}\right)} \right),
$$
(6.3)

where $P_n \left(\frac{\ell}{L}\right)$ denotes the an $n$th order polynomial of $\frac{\ell}{L}$. The coefficients of the polynomials can depend on the initial data $\epsilon_0, p, Q, \bar{Q}$ and the (generalized) background fields $a, \bar{a}, b, L$. We require that as $\frac{\ell}{L} \to 0$ we recover the initial condition given in (4.26). By matching to a high order series solution of (4.23), the Ansatz (6.3) can be verified and the coefficients determined.

The string construction to be discussed in Section 6.4 suggests additional structure, namely that $E_n$ is a solution of a quadratic equation. Following this hint, the most compact

form that we managed to bring the solution to is:

$$\epsilon_n \equiv E_n \hat{L} = s + \frac{-B - \sqrt{B^2 - 4AC}}{2A}, \qquad 0 = A(\epsilon_n - s)^2 + B(\epsilon_n - s) + C,$$

$$\hat{L} \equiv (1 - b^2)L, \quad \mu \equiv \frac{\pi \ell}{\hat{L}}, \quad \hat{Q} \equiv Q + 2\pi aL, \quad \hat{\bar{Q}} \equiv \bar{Q} + 2\pi \bar{a}L, \quad \mathcal{A} \equiv aL \quad \bar{\mathcal{A}} \equiv \bar{a}L,$$

$$G_{J\bar{T}} \equiv (1 - b)g_{J\bar{T}}, \quad G_{J\Theta} \equiv (1 + b)g_{J\Theta}, \quad G_{\bar{J}T} \equiv (1 + b)g_{\bar{J}T}, \quad G_{\bar{J}\bar{\Theta}} \equiv (1 - b)g_{\bar{J}\bar{\Theta}},$$

$$\hat{G}_{T\bar{T}} \equiv G_{T\bar{T}} + \frac{1}{2}\pi(G_{J\bar{T}}G_{J\Theta} + G_{\bar{J}T}G_{\bar{J}\bar{\Theta}}), \quad G_{T\bar{T}} \equiv (1 - b^2)g_{T\bar{T}},$$

$$s = \frac{1 + b}{2}\epsilon + \frac{1 - b}{2}\bar{\epsilon}, \quad \epsilon \equiv \epsilon_0 + p + 2\mathcal{A}(\hat{Q} - \pi\mathcal{A}), \quad \bar{\epsilon} \equiv \epsilon_0 - p + 2\bar{\mathcal{A}}(\hat{\bar{Q}} - \pi\bar{\mathcal{A}}),$$

$$A = \left(\frac{\pi}{2}(G_{J\bar{T}}^2 + G_{\bar{J}T}^2) + \hat{G}_{T\bar{T}}\right)\mu^2,$$

$$B = -1 - (G_{J\bar{T}}\hat{Q} + G_{\bar{J}T}\hat{\bar{Q}})\mu + \left((\pi G_{\bar{J}T}^2 + \hat{G}_{T\bar{T}})\epsilon + (\pi G_{J\bar{T}}^2 + \hat{G}_{T\bar{T}})\bar{\epsilon}\right)\mu^2,$$

$$C = -\left(G_{\bar{J}T}\hat{\bar{Q}}\epsilon + G_{J\bar{T}}\hat{Q}\bar{\epsilon}\right)\mu + \left(\frac{\pi}{2}G_{\bar{J}T}^2\epsilon^2 + \hat{G}_{T\bar{T}}\epsilon\bar{\epsilon} + \frac{\pi}{2}G_{J\bar{T}}^2\bar{\epsilon}^2\right)\mu^2.$$

$$(6.4)$$

Let us highlight some properties of this lengthy set of expressions. When $\mu = 0$, corresponding to only turning on background gauge fields in a CFT, $A = C = 0$ and $B = -1$, hence $\epsilon_n = s$, which matches (4.26) once we account for the fact that $E_n = \epsilon_n/\hat{L}$; of course, this was the initial condition that we used to find the solution given in (6.4). It is not surprising that by forming dimensionless combinations $\mu, \mathcal{A}, \bar{\mathcal{A}}$ we simplified formulas. Curiously, we managed to absorb all the $b$-dependence of $A, B, C$ into the redefined couplings $G_{\mathcal{O}}$ and $\mu$, but even using these definitions $s$ still depends on $b$ explicitly. We also managed to absorb all the $G_{J\Theta}, G_{\bar{J}\bar{\Theta}}$ dependence into a shifted $T\bar{T}$ coupling, $\hat{G}_{T\bar{T}}$ defined in (6.4). That $A, B, C, s$ are polynomials in $\epsilon_0, p, \mathcal{A}, \bar{\mathcal{A}}, \hat{Q}, \hat{\bar{Q}}$ is a consequence of (4.14) truncating at finite order.

We used the background fields mainly as an auxiliary device. If we are only interested in bilinear deformations, we can turn the background fields off: $\mathcal{A} = \bar{\mathcal{A}} = b = 0$, $\hat{Q} = Q$, $\hat{\bar{Q}} = \bar{Q}$, $G_{\mathcal{O}} = g_{\mathcal{O}}$. In the absence of background fields, the expressions simplify significantly: $\epsilon = \epsilon_0 + p$ and $\bar{\epsilon} = \epsilon_0 - p$. The special cases of $J\bar{T}$ and $T\bar{T}$ deformations given in (6.1) and (6.2) are reproduced as special cases.

For large initial energy $\epsilon_0$ with other quantum numbers fixed, the spectrum formally behaves as

$$\epsilon_n = \sqrt{\left(-\frac{\epsilon_0}{A} + O(1)\right)} + O(1),$$

$$A = \left(\frac{1}{2}\pi(G_{J\bar{T}}^2 + G_{\bar{J}T}^2) + G_{T\bar{T}} + \frac{1}{2}\pi(G_{J\bar{T}}G_{J\Theta} + G_{\bar{J}T}G_{\bar{J}\bar{\Theta}})\right)\mu^2,$$

$$(6.5)$$

where we repeated $A$ from (6.4) for convenience. For $A > 0$, states with large initial energies become complex for some value of $\mu$. In [14] it was proposed for the $T\bar{T}$ deformation that such states should be discarded from the spectrum, turning the theory into a quantum mechanical theory with a finite number of states. The validity of this proposal is clearly beyond what can be assessed by the local field theory tools used in this paper. For $A < 0$ the energies are real, and combined with the Cardy growth of the density of states this leads to Hagedorn growth of the density of states [8, 15]. For the pure $J\bar{T}$ and $\bar{J}T$ deformations, one always has $A > 0$, and the spectrum necessarily becomes complex [5]. Once we turn on $T\bar{T}$ (or equivalently $J\Theta$ in the presence of $J\bar{T}$) with a sufficiently negative coupling constant, we can make $A < 0$ and the asymptotic spectrum real. It may be interesting to

study the $A \to 0^-$ limit: the spectrum appears to remain real and bounded below, with some states acquiring infinite energy.[11]

Finally, it is natural to ask about the meaning of the other branch of the square root in (6.4), $\epsilon_n^{(+)} = s + \frac{-B+\sqrt{B^2-4AC}}{2A}$. For $\mu \to 0$ the energy of these states would diverge. Nevertheless, these "eigenvalues" play an interesting role in the modular differential equation that the torus partition function obeys in $T\bar{T}$ and $J\bar{T}$ deformed theories [18, 19].

## 6.2 Exploring the coupling space

To explore some properties of the spectrum, we ask what happens if we turn on the deformations one after another instead of simultaneously, which gives (6.4). See Figure 2 for a sketch of the situation. We will only work with two couplings and turn off all the others. We leave the exploration of more complicated paths in coupling space for future work.

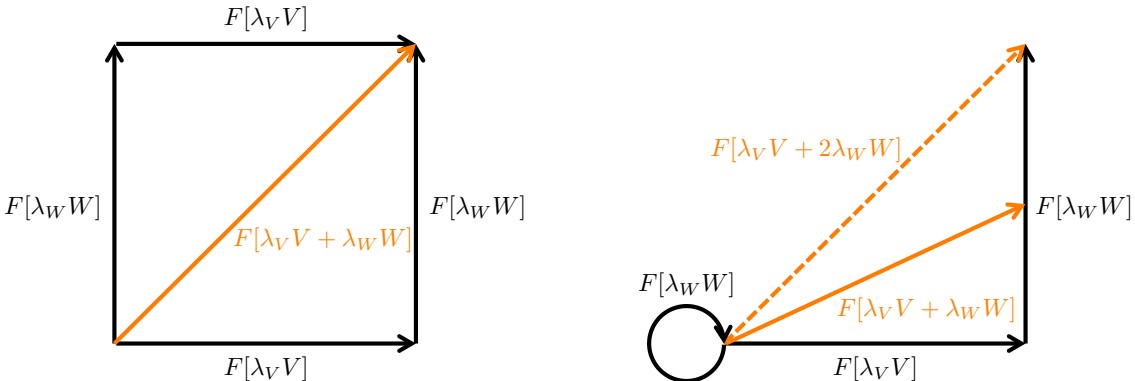

Figure 2: **Left:** Graphical representation of (6.7). Independent of which order we evolve the spectrum of the CFT with $V$ and $W$, we get the same spectrum. The result also agrees by the simultaneous irrelevant deformation by $V$ and $W$ represented by the diagonal orange arrow. **Right:** For the special case of $V = J\bar{T}$, $W = J\Theta$, the structure of the coupling space is more complicated, as explained in (6.6).

Let us turn on $\lambda_V$ first, where $V$ is one of the five irrelevant operators that we are studying and $\lambda_V$ is the dimensionful version of $g_V$, see (4.22). The spectrum with background gauge fields turned on is obtained by setting $g_{\mathcal{O}} = 0$, $(\mathcal{O} \neq V)$ in (6.4). We can use this result as initial condition for the $\lambda_W$, $(W \neq V)$ flow equations for the spectrum given in Table 3. This is a more complicated initial condition than the conformal initial conditions (4.23). Nevertheless, the flow is still solvable in a closed form. Setting the background fields to zero at the end, we obtain the spectrum of the theory first deformed by $V$ and then by $W$, which we denote by $F[\lambda_W W]F[\lambda_V V]\sigma_{CFT}$, where $\sigma_{CFT}$ is the CFT spectrum and $F[\lambda_{\mathcal{O}}\mathcal{O}]$ is a symbolic operator implementing the flow. We find that the only nontrivial flow is obtained for $V = J\bar{T}$, $W = J\Theta$ (and similarly for their conjugates), for which

$$
\begin{aligned}
F[\lambda_{J\bar{T}}\, J\bar{T}]F[\lambda_{J\Theta}\, J\Theta]\sigma_{CFT} &= F[\lambda_{J\bar{T}}\, J\bar{T}]\sigma_{CFT}\,, \\
F[\lambda_{J\Theta}\, J\Theta]F[\lambda_{J\bar{T}}\, J\bar{T}]\sigma_{CFT} &= F[\lambda_{J\bar{T}}\, J\bar{T} + 2\lambda_{J\Theta}\, J\Theta]\sigma_{CFT}\,.
\end{aligned}
\tag{6.6}
$$

The first equation is easy to understand: in a CFT $J\Theta = 0$, hence $F[\lambda_{J\Theta}]\sigma_{CFT} = \sigma_{CFT}$. The second equation is the result of a nontrivial computation. By $F[\lambda_V V + \lambda_W W]$ we mean

---

[11]We thank David Kutasov and Soumangsu Chakraborty for discussing some upcoming work.

861 the specific flow that led to (6.4), i.e. we use the common scale $\ell$ as defined in (4.22) and flow
862 with the combined flow equation (4.23). Note the factor of 2 multiplying $\lambda_{J\Theta}$ in the second
863 equation. Because (6.4) only depends on $\hat{G}_{T\bar{T}}$, which is a linear combination of $\lambda_{J\bar{T}}\lambda_{J\Theta}$ and
864 $\lambda_{T\bar{T}}$, we could have written $F[\lambda_{J\bar{T}}\,J\bar{T}+2\lambda_{J\Theta}\,J\Theta]$ equivalently as $F[\lambda_{J\bar{T}}\,J\bar{T}+\#\,J\Theta+\#T\bar{T}]$.
865 For any other pair of operators:

$$F[\lambda_W W]F[\lambda_V V]\sigma_{CFT} = F[\lambda_W W]F[\lambda_V V]\sigma_{CFT} = F[\lambda_V V + \lambda_W W]\sigma_{CFT}\,. \qquad (6.7)$$

866 We conclude that the structure of the coupling space is rather simple, as most deforma-
867 tions commute. In particular, making two deformation in succession does not lead out of
868 the space of theories that we can reach by the simultaneous deformation by all operators,
869 as given in (6.4).

## 6.3 Solving and checking the $J\bar{J}$ deformation

871 Recall that because we were using a dimensionful parameter to control the flow, we did not
872 cover the case of the $J\bar{J}$ deformation, when solving (4.23). The solution of the differential
873 equation given in the first row of Table 3 is a lot simpler than that of (4.23). The
874 introduction of the same variables as in (6.4) is useful, and we get:

$$\epsilon_n = E_n \hat{L} = s + \hat{Q}\hat{\bar{Q}}\sinh(2\pi\lambda_{J\bar{J}}) + (\hat{Q}^2 + \hat{\bar{Q}}^2)\sinh^2(\pi\lambda_{J\bar{J}})\,. \qquad (6.8)$$

875 In [51] the change in the scaling dimension of certain primary operators was obtained
876 using AdS/CFT and confirmed to second order in perturbation theory. The result in their
877 equation (5.1), after redefining $q = \sqrt{\frac{k}{2}}Q$, $\tilde{q} = \sqrt{\frac{\tilde{k}}{2}}\bar{Q}$, $h \equiv \frac{2H}{\pi\sqrt{k\tilde{k}}}$, reads

$$\epsilon_n = \epsilon_0 - \frac{2H}{1-H^2}Q\bar{Q} + \frac{H^2}{1-H^2}(Q^2 + \bar{Q}^2)\,. \qquad (6.9)$$

878 Because $H$ and $\lambda_{J\bar{J}}$ are dimensionless, and the space of theories is one dimensional, different
879 definitions can give different parametrizations of the same line. Indeed setting

$$\lambda_{J\bar{J}} = -\frac{1}{2\pi}\text{arcsinh}\left(\frac{2H}{1-H^2}\right) \qquad (6.10)$$

880 in (6.8) and setting the background fields to zero produces (6.9). This is another nice check
881 of the validity of our formalism.

## 6.4 A check from string theory

883 In this section we use the string construction of [5,15,34] to test a special case of the energy
884 formula (6.3), where the background fields are set to zero $\mathcal{A} = \bar{\mathcal{A}} = b = 0$ and the $J\Theta$ and
885 $\bar{J}\bar{\Theta}$ deformations are turned off. We obtain a precise match.
886 We now give a lightning review of the argument of [5], skipping over many important
887 details. Let us consider Type II superstrings on the background (massless BTZ)$\times S^1 \times \mathcal{N}$.
888 Vertex operators of the worldsheet theory dual to certain Ramond sector states of the dual
889 CFT$_2$ were constructed in [5], whose explicit form we will not need. The construction uses
890 separate primaries in the (massless BTZ)$\times S^1$ and in the $\mathcal{N}$ CFTs. The Virasoro constraint
891 imposes:

$$\begin{aligned} 0 &= \Delta_1 + \Delta_2 - \frac{1}{2}\,, \\ 0 &= \bar{\Delta}_1 + \bar{\Delta}_2 - \frac{1}{2}\,, \end{aligned} \qquad (6.11)$$

892  where $\Delta_{1,2}$ are the left scaling dimensions in the (massless BTZ)$\times S^1$ and $\mathcal{N}$ CFTs respec-
893  tively. To simplify formulas, we restrict to the winding number 1 sector of the theory. The
894  scaling dimensions in the (massless BTZ)$\times S^1$ CFT are given by:

$$
\begin{aligned}
\Delta_1 &= -E_L + \frac{Q^2}{8\pi^2} - \frac{j(j+1)}{k}\,, \\
\bar{\Delta}_1 &= -E_R + \frac{\bar{Q}^2}{8\pi^2} - \frac{j(j+1)}{k}\,, \\
E_{L,R} &\equiv \frac{1}{2}(E \pm P)\,,
\end{aligned}
\tag{6.12}
$$

895  where $j$ is a quantum number related to radial motion, $k = \left(\frac{L_{\text{AdS}}}{\ell_s}\right)^2$, and we set $R =$
896  $1$, $q_L = \frac{Q}{2\pi}$, $q_R = \frac{\bar{Q}}{2\pi}$ in the formulas of [5].[12]
897     The (massless BTZ)$\times S^1$ CFT is an $SL(2,\mathbb{R})_k \times U(1)$ WZW model, and hence has
898  interesting exactly marginal $J\bar{J}$ deformations. It was argued in [5,15] that a deformation
899  that to linear order agrees with the $J^-_{SL}\bar{J}^-_{SL}$ deformation in the terminology of this paper is
900  related to a single trace version of the $T\bar{T}$ deformation of the dual CFT, while the $J_{U(1)}\bar{J}^-_{SL}$
901  and $J^-_{SL}\bar{J}_{U(1)}$ deformation is related to the $J\bar{T}$ and $\bar{J}T$ deformations [5]. The arguments
902  are complicated and rely on some conjectures about the dual CFT; the proposal is that the
903  string states created by the vertex operators discussed above should evolve in the same
904  way under the single trace deformation version of the irrelevant deformations as they would
905  under their double trace versions to which our field theory treatment applies.
906     Under the $J\bar{J}$ deformations of the (massless BTZ)$\times S^1$ CFT the scaling dimensions in
907  the $\mathcal{N}$ CFT do not change, while the change of $\Delta_1$ and $\bar{\Delta}_1$ can be determined by combining
908  formulas (5.19)-(5.22), (5.29), and (5.32)-(5.33) from [5]. Adapting their equations to our
909  notation, which includes introducing the coupling constants $g_{J\bar{T}}$, $g_{\bar{J}T}$, $\tilde{g}_{T\bar{T}}$ with appropriate
910  numerical prefactors, gives[13]

$$
\begin{aligned}
\Delta_1 &= -E_L + \frac{Q^2}{8\pi^2} - g_{\bar{J}T}\mu\bar{Q}E_L + 2\pi^2 g_{\bar{J}T}^2\mu^2 E_L^2 \\
&\quad - g_{J\bar{T}}\mu Q E_R + 2\pi^2 g_{J\bar{T}}^2\mu^2 E_R^2 + 4\pi(\tilde{g}_{T\bar{T}} - \pi g_{J\bar{T}}g_{\bar{J}T})\mu^2 E_L E_R - \frac{j(j+1)}{k}\,, \\
\bar{\Delta}_1 &= -E_R + \frac{\bar{Q}^2}{8\pi^2} - g_{\bar{J}T}\mu\bar{Q}E_L + 2\pi^2 g_{\bar{J}T}^2\mu^2 E_L^2 \\
&\quad - g_{J\bar{T}}\mu Q E_R + 2\pi^2 g_{J\bar{T}}^2\mu^2 E_R^2 + 4\pi(\tilde{g}_{T\bar{T}} - \pi g_{J\bar{T}}g_{\bar{J}T})\mu^2 E_L E_R - \frac{j(j+1)}{k}\,.
\end{aligned}
\tag{6.13}
$$

911  Subtracting the two equations leads to a $\mu$ independent result, which expresses that the spin
912  of the vertex operator is quantized. Henceforth, we drop the second equation. We remark
913  that the field theory explanation of why we can simply add together the contributions of
914  different deforming operators is that these deformations commute in the sense explained in
915  Section 6.2. We also note that linearizing in the couplings $g_{J\bar{T}}$, $g_{\bar{J}T}$, $\tilde{g}_{T\bar{T}}$, we get

$$
\delta E = \delta(\Delta_1 + \bar{\Delta}_1) = -2g_{J\bar{T}}Q(\mu E_R) - 2g_{\bar{J}T}\bar{Q}(\mu E_L) + 4\pi\tilde{g}_{T\bar{T}}(\mu E_L)(\mu E_R)\,,
\tag{6.14}
$$

916  which to linear order agrees with the spectrum of the $J\bar{J}$-deformed theory (6.8), if we identify
917  the charges of $J_{U(1)}$, $\bar{J}_{U(1)}$, $J^-_{SL}$, $\bar{J}^-_{SL}$ with $Q$, $\bar{Q}$, $(\mu E_L)$, $(\mu E_R)$, and $g_{J\bar{T}}$, $g_{\bar{J}T}$, $g_{T\bar{T}}$ with
918  (up to constant factors) the coupling $\lambda_{J\bar{J}}$ of the $J\bar{J}$ operators $J_{U(1)}\bar{J}^-_{SL}$, $J^-_{SL}\bar{J}_{U(1)}$, $J^-_{SL}\bar{J}^-_{SL}$.

---

[12]A check on these normalization factors is that if we set $r = \sqrt{2}$ in (B.9), we get the same spectrum of scaling dimensions, as in [5].

[13]In the first version of the paper we were missing the shift of $\tilde{g}_{T\bar{T}}$ by $-\pi g_{J\bar{T}}g_{\bar{J}T}$. We thank Soumangsu Chakraborty and Amit Giveon for pointing this out to us.

To higher orders the agreement does not hold, demonstrating that the worldsheet CFT is not precisely a $J\bar{J}$ deformation of the (massless BTZ)$\times S^1$ CFT in the terminology of this paper.

To get the evolution of energies in the boundary theory, we put $(0)$ superscripts on the quantities in the undeformed theory given in (6.12). The Virasoro constraint (6.11) implies that we have to equate $\Delta_1$ (and $\bar{\Delta}_1$) in (6.12) and (6.13), which then implies:

$$-E_L^{(0)} + \frac{Q^2}{8\pi^2} = -E_L + \frac{Q^2}{8\pi^2} - g_{\bar{J}T}\mu\bar{Q}E_L + 2\pi^2 g_{\bar{J}T}^2\mu^2 E_L^2$$
$$- g_{J\bar{T}}\mu Q E_R + 2\pi^2 g_{J\bar{T}}^2\mu^2 E_R^2 + 4\pi(\tilde{g}_{T\bar{T}} - \pi g_{J\bar{T}}g_{\bar{J}T})\mu^2 E_L E_R\,. \tag{6.15}$$

If we define the $T\bar{T}$ coupling as $g_{T\bar{T}} \equiv \tilde{g}_{T\bar{T}} - \pi g_{J\bar{T}}g_{\bar{J}T}$, then the above equation can be brought to the form:

$$0 = (2\pi)^2 A(E - E^{(0)})^2 + 2\pi B(E - E^{(0)}) + C\,, \tag{6.16}$$

which is equivalent to (6.4) with all the background fields turned off, $\mathcal{A} = \bar{\mathcal{A}} = b = 0$, $\hat{Q} = Q$, $\hat{\bar{Q}} = \bar{Q}$, $G_{\mathcal{O}} = g_{\mathcal{O}}$, and with the $J\Theta$ and $\bar{J}\bar{\Theta}$ deformations turned off, $g_{J\Theta} = g_{\bar{J}\bar{\Theta}} = 0$. The factors of $2\pi$ account for the fact that we set $L = 2\pi$ by setting $R = 1$, hence $\epsilon = 2\pi E$. (Also $p = 2\pi P$.)

# 7  Conclusions and outlook

In this paper, we presented detailed arguments for the proposal of the energy spectrum of CFTs deformed simultaneously by $J\bar{T}$, $J\Theta$, $\bar{J}T$, $\bar{J}\bar{\Theta}$, $T\bar{T}$ and also by the current components $J_x$, $\bar{J}_x$, $T_{tx}$ which are equivalent to turning on the background fields $a$, $\bar{a}$, $b$ in (6.4). Note that deforming by the time component of conserved currents would be trivial: since the corresponding charges commute with the Hamiltonian, the eigenvalues $Q$, $\bar{Q}$, $E$, $P$ would simply add under such deformations.

We have arrived at this spectrum following the strategy outlined in Section 2, see also Figure 1. We implemented the first step of the strategy, rigorously determining the initial conditions for the flow equation for the Hamiltonian. We then used the example of the classical free scalar to conjecture a universal equation valid at an arbitrary point in coupling space in (4.18) and Table 2. We performed two checks of this equation in Section 5: we checked a special case of the equation in a more general classical field theory setting in Section 5.1, and a quantum mechanical check in low order perturbation theory in Section 5.2. It would be interesting to go to higher orders in perturbation theory. Ultimately, in the future we would like to find a nonperturbative quantum derivation of these equations. From these universal equations, the flow equation for the energy follows using the factorization property of the special composite operators discussed in this paper; this is again a fully rigorous step. We then solved this equation using the initial conditions to obtain the spectrum (6.4). That we have obtained elegant solvable equations for the spectrum, whose solution reproduces previously known special cases gives us confidence that this is the full quantum answer. We solved the $J\bar{J}$ deformation with the same methods, reproducing the spectrum found using conformal perturbation theory and AdS/CFT in [51]. We also did a new AdS/CFT computation in Section 6.4 of the spectrum of the $J\bar{T}$, $\bar{J}T$, $T\bar{T}$ deformed theory with the background fields turned off, and this confirmed the spectrum (6.4) in a special case. We provided evidence in Section 6.2 that turning on the couplings of the irrelevant operators in different order (instead of simultaneously) will not lead out of the space of theories we solved.

This work leaves many interesting directions open for future investigation. It would be interesting to obtain the spectrum of non-conformal theories deformed by irrelevant operators. While the universal operator equations and hence the flow equations for the spectrum apply, we do not know how to obtain the initial conditions. An exception is provided by Gaussian theories, the massive complex boson and fermion, where turning on background gauge fields preserves their Gaussian nature and hence solvability. We leave their solution to future work. The universal equations for the Hamiltonian can be solved for the classical Hamiltonian (and Lagrangian). We have only discussed this in Appendix C, since we do not know how to quantize these theories starting from the Lagrangian. Since we do know how to understand these theories using flow equations, this could give insight into how to treat such exotic theories that include the Nambu-Goto string (in static gauge) [3,8]. We would also like to understand how to turn on the background fields in the AdS/CFT setup analyzed in Section 6.4. It would be interesting to analyze the torus partition function of this class of theories, and fascinating to prove uniqueness results similar to those found in [18,19].

The most interesting extension of this work would be to understand deformations by bilinear composite operators built from the higher spin KdV currents. Understanding these would presumably lead to qualitatively new UV behaviors. In Section 3.4 we attempted to obtain the initial conditions for this flow, and explained why our approach does not apply straightforwardly. Despite this failure, it would be interesting to understand whether there exists a universal operator equation governing the Hamiltonian at an arbitrary point in coupling space. Since the background field for the spin $s$ current is irrelevant in this case, $[\alpha_s] = 2 - s$, we expect a proliferation of terms. A first step in this direction would be to work out the case of the classical free scalar. We also note that we take a step in a tanglential direction in a future publication, where we compute the spectrum of KdV conserved charges in the $T\bar{T}$ flow.

# Acknowledgements

We thank Costas Bachas, Shouvik Datta, Guido Festuccia, Yunfeng Jiang, David Kutasov, Philippe LeFloch, Stefano Negro, Ilia Smilga, Tin Sulejmanpasic, Jan Troost, Herman Verlinde, Yifan Wang for useful discussions and Amit Giveon for coordinating the arXiv submission of [50] with our paper. MM is supported by the Simons Center for Geometry and Physics. BLF gratefully acknowledges support from the Simons Center for Geometry and Physics, Stony Brook University at which some of the research for this paper was performed.

# A   Conventions

We use the following conventions. The complex coordinates in Euclidean space are defined by:

$$z = x + it, \qquad \bar{z} = x - it. \tag{A.1}$$

The cylinder $S^1 \times \mathbb{R}$ has circumference $L$, hence $z \sim z + L$. The stress tensor in a relativistic theory is defined by

$$T_{\mu\nu} \equiv \frac{2}{\sqrt{g}} \frac{\delta S}{\delta g^{\mu\nu}}. \tag{A.2}$$

We will be dealing with theories that cannot be easily coupled to gravity, as they do not have Lorentz invariance, and for these we will be using the Noether stress tensor that obeys:

$$0 = \partial^\nu T_{\mu\nu} \,, \tag{A.3}$$

with the corresponding (Hermitian) conserved quantities:

$$H = -\int_0^L dx \, T_{tt} \,, \qquad P = -i \int_0^L dx \, T_{xt} \,. \tag{A.4}$$

They generate the spacetime translations:

$$[H, \mathcal{O}] = \partial_t \mathcal{O} \,, \qquad [P, \mathcal{O}] = i\partial_x \mathcal{O} \,. \tag{A.5}$$

One useful example to keep in mind is that for a scalar theory whose Lagrangian $\mathcal{L}(\phi, \partial_\mu \phi)$ does not contain higher derivatives (but may contain arbitrary powers of single derivatives), we have

$$T_{\mu\nu} \equiv \frac{\partial \mathcal{L}}{\partial(\partial^\nu \phi)} \partial_\mu \phi - \delta_{\mu\nu} \mathcal{L} \,, \tag{A.6}$$

which obeys (A.3) by the equations of motion.

In CFT the components of the stress tensor are usually denoted by

$$T \equiv -2\pi T_{zz} \,, \qquad \Theta \equiv +2\pi T_{z\bar{z}} \,, \qquad \bar{\Theta} \equiv +2\pi T_{\bar{z}z} \,, \qquad \bar{T} \equiv -2\pi T_{\bar{z}\bar{z}} \,, \tag{A.7}$$

and the conservation equation written as

$$\bar{\partial}T = \partial\Theta \,, \qquad \partial\bar{T} = \partial\bar{\Theta} \,. \tag{A.8}$$

For convenience, we also write down the components of the stress tensor in $(x, t)$ coordinates:

$$
\begin{aligned}
T_{tt} &= \frac{1}{2\pi} \left( T + \Theta + \bar{\Theta} + \bar{T} \right) \,, & T_{tx} &= -\frac{i}{2\pi} \left( T - \Theta + \bar{\Theta} - \bar{T} \right) \,, \\
T_{xt} &= -\frac{i}{2\pi} \left( T + \Theta - \bar{\Theta} - \bar{T} \right) \,, & T_{xx} &= -\frac{1}{2\pi} \left( T - \Theta - \bar{\Theta} + \bar{T} \right) \,.
\end{aligned}
\tag{A.9}
$$

For a conserved current corresponding to an internal symmetry, we have

$$
\begin{aligned}
0 &= \partial^\mu J_\mu \\
Q &= \int_0^L dx \, J_t(x)
\end{aligned}
\tag{A.10}
$$

From the perspective of Euclidean field theory a natural formula would instead be $-i \int_0^L dx \, J_t(x)$, because we want $Q$ to be Hermitian, and to Wick-rotate an operator with spin, we conventionally add factors of $i$ to the time components. However, to conform with CFT convention, we chose to omit the $i$ from (A.10). If we have only one current a natural choice for normalization is that the charge is an integer. However, as we recall on the example of the free compact boson in Appendix B, this normalization is not always natural. In CFT, it is customary to use the notation $J \equiv J_z, \bar{J} \equiv \bar{J}_{\bar{z}}$.

## B   Free compact boson

We take the free massless scalar Lagrangian to be:

$$\mathcal{L} = \frac{1}{8\pi}(\partial_\mu\phi)^2\,. \tag{B.1}$$

The equation of motion is:

$$0 = (\partial_t^2 + \partial_x^2)\phi\,. \tag{B.2}$$

The propagator is

$$\langle\phi(z)\phi(0)\rangle = -\log|z|^2\,. \tag{B.3}$$

The currents and the stress tensor (on the plane) are given by:

$$
\begin{aligned}
J &= i\partial\phi\,, \qquad \bar{J} = -i\bar{\partial}\phi\,, \\
T &= -\frac{1}{2}:(\partial\phi)^2: = \frac{1}{2}:J^2:\,, \qquad \bar{T} = -\frac{1}{2}:(\bar{\partial}\phi)^2: = \frac{1}{2}:\bar{J}^2:\,.
\end{aligned}
\tag{B.4}
$$

The canonical momentum and Hamiltonian densities are defined by

$$
\begin{aligned}
\Pi &\equiv i\frac{\partial L}{\partial(\partial_t\phi)} = \frac{i}{4\pi}\partial_t\phi \\
\mathcal{H} &\equiv i\Pi\partial_t\phi + \mathcal{L} = 2\pi\Pi^2 + \frac{1}{8\pi}(\partial_x\phi)^2\,.
\end{aligned}
\tag{B.5}
$$

The canonical commutation relations are $[\Pi(x),\phi(y)] = -i\delta(x-y)$. It is easy to check that $\mathcal{H} = -\frac{1}{2\pi}\left(T + \bar{T}\right)$, consistent with (A.4) and (A.9). Hamilton's equations are:

$$
\begin{aligned}
\partial_t\Pi &= i\frac{\delta H}{\delta\phi} = -i\partial_x\left(\frac{\partial\mathcal{H}}{\partial(\partial_x\phi)}\right)\,, \\
\partial_t\phi &= -i\frac{\delta H}{\delta\Pi} = -i\left(\frac{\partial\mathcal{H}}{\partial\Pi}\right)\,,
\end{aligned}
\tag{B.6}
$$

which are consistent with (B.2). Since the Hamiltonian is the same as in usual Lorentzian quantum mechanics, (B.6) are just the usual Lorentzian Hamilton's equations with the replacement $\frac{\partial}{\partial t_L} = i\frac{\partial}{\partial t}$.

We take the boson to be compact with radius $r$:

$$\phi \sim \phi + 2\pi r\,. \tag{B.7}$$

The mode expansion of the scalar and the currents on the cylinder is:

$$
\begin{aligned}
\phi &= \phi_0 + \frac{4\pi n}{rL}(-it) - \frac{2\pi rw}{L}x + \text{(oscillating terms)} \\
J &= i\partial\phi = -\frac{i}{2}\left(\frac{4\pi n}{rL} + \frac{2\pi rw}{L}\right) + \text{(oscillating terms)} \\
\bar{J} &= -i\bar{\partial}\phi = \frac{i}{2}\left(\frac{4\pi n}{rL} - \frac{2\pi rw}{L}\right) + \text{(oscillating terms)}\,.
\end{aligned}
\tag{B.8}
$$

Then the charges using (A.10) are:

$$Q = \frac{2\pi n}{r} + \pi rw\,, \qquad \bar{Q} = \frac{2\pi n}{r} - \pi rw\,. \tag{B.9}$$

1033  The charges are quantized, but are not integers. In fact, for irrational $r^2$, there does not
1034  exist a normalization in which they would be integer valued.

1035      At $\lambda = 0$, solving the differential equations (4.3) and (4.4) with the expressions for the
1036  conserved current components given in (4.2) gives:

$$\mathcal{H} = \frac{2\pi \left(\Pi + \frac{a-\bar{a}}{2}\right)^2 + \frac{1}{8\pi}\left(\partial_x\phi - 2\pi(a+\bar{a})\right)^2 - b\left(\Pi + \frac{a-\bar{a}}{2}\right)\left(\partial_x\phi - 2\pi(a+\bar{a})\right)}{1 - b^2}. \quad \text{(B.10)}$$

1037  The shifts of $\Pi$ and $\partial_x\phi$ are explained by the comment around (4.3).

# C  Hamiltonian and Lagrangian of the deformed free scalar

1039  We have not determined the closed form solution of the flow equations for the Hamiltonian
1040  density, (4.18) and Table 2. They can be recovered from the solution of the spectrum (6.4)
1041  using the following very simple recipe. In $\epsilon_n(\epsilon_0, p, Q, \bar{Q})$ as a function of the four initial
1042  conditions, we have to make the replacements:

$$\mathcal{H} = \frac{1}{(1-b^2)L^2} \epsilon_n \left( L^2 \left( 2\pi\Pi^2 + \frac{1}{8\pi}(\partial_x\phi)^2 \right), -L^2\,\Pi\partial_x\phi, -\frac{L}{2}(\partial_x\phi - 4\pi\Pi), -\frac{L}{2}(\partial_x\phi + 4\pi\Pi) \right). \quad \text{(C.1)}$$

1043  An intuitive way to obtain this formula is to take a simple classical phase space configuration,
1044  for which

$$\partial_x\phi = -\frac{2\pi rw}{L}, \qquad \Pi = \frac{n}{rL},$$
$$\epsilon_0 = 2\pi\left(\frac{n^2}{r^2} + \frac{r^2w^2}{4}\right), \qquad p = 2\pi nw, \qquad Q = \frac{2\pi n}{r} + \pi rw, \qquad \bar{Q} = \frac{2\pi n}{r} - \pi rw. \quad \text{(C.2)}$$

1045  In the undeformed case, the appropriate field configuration of $\phi$ is given in (B.8) (with the
1046  oscillating terms set to zero), but after deformation $\Pi \neq \frac{i}{4\pi}\partial_t\phi$. Plugging these into $\mathcal{H}$
1047  defined by (C.1) and multiplying by $L$ to perform the trivial integral over $x$, we recover
1048  the spectrum (6.4). This is not quite a proof of (C.1), since for the special configurations
1049  in (C.2) $\epsilon_0$ and $p$ are determined by $Q, \bar{Q}$. Nevertheless, it can be checked explicitly that
1050  $\mathcal{H}$ resulting from (C.1) indeed solves the appropriate equations. We note, that (B.10)
1051  can indeed be recovered by using (C.1), with $\epsilon_n$ replaced by the initial condition s form
1052  (6.4). Finally, the Lagrangian for the deformed theories can be obtained by Legendre
1053  transformation,

$$\frac{\partial\mathcal{H}}{\partial\Pi} = i\partial_t\phi,$$
$$\mathcal{L} = -i\Pi\,\partial_t\phi + \mathcal{H}. \quad \text{(C.3)}$$

1054      Conversely, given a Lagrangian of a shift symmetric scalar field $\mathcal{L}(\partial\phi, \bar{\partial}\phi)$, we can
1055  conjecture its energy spectrum by first going to the Hamiltonian $\mathcal{H}(\partial_x\phi, \Pi)$, plugging in the
1056  first line of (C.2), and expressing $n, w$ with the possible initial conditions, $\epsilon_0, p, Q, \bar{Q}$. As
1057  already mentioned above, since there are four initial conditions and only the two $n, w$ in the
1058  output of this procedure, obtaining the right spectrum this way requires some guesswork.
1059  In the case of the $T\bar{T}$ deformation (with background fields turned off), the spectrum cannot
1060  depend on $Q, \bar{Q}$ and one obtains the correct spectrum from the Lagrangian

$$\mathcal{L}_{T\bar{T}} = \frac{1}{2\pi^2\ell^2}\left(1 - \sqrt{1 - \frac{\pi\ell^2}{2}(\partial_\mu\phi)^2}\right) \quad \text{(C.4)}$$

1061  as was shown in [22] using the method described here.

## 1062 D   Comments on the $J\bar{T}$ deformation

1063 At first sight, it may seem that our definition of the $J\bar{T}$ deformation and the one in the
1064 literature is different. We have been working with a non-holomorphic current that had
1065 quantized charge that does not change under the deformation, while in the literature the
1066 current $J$ is holomorphic and its charge depends on the scale $\ell$ [5]. It turns out that the
1067 two definitions of the theories are equivalent, as we explain below.

1068     Let us determine the explicit form of $\mathcal{H}$ for the case of the $J\bar{T}$ deformation with the
1069 background fields turned off. We can use the simple explicit formula given in (6.1) instead
1070 of (6.4). Plugging into (C.1) we obtain:

$$
\begin{aligned}
\mathcal{H}_{J\bar{T}} &= \frac{1 - \frac{\pi\ell}{2}(\partial_x\phi - 4\pi\Pi) - \pi^3\ell^2\Pi\partial_x\phi - \sqrt{(1 + 4\pi^2\ell\Pi)(1 - \pi\ell\partial_x\phi)}}{\pi^3\ell^2}\,, \\
\mathcal{L}_{J\bar{T}} &= \frac{1}{2\pi}\partial\phi\frac{\bar{\partial}\phi}{1 - \pi\ell\bar{\partial}\phi}\,.
\end{aligned}
\tag{D.1}
$$

1071 The latter Lagrangian was obtained in [5,6]. It can be checked that the current

$$
\hat{J}_\mu \equiv J_\mu - 2\pi^2 i\ell\, T_{\bar{z}\mu}
\tag{D.2}
$$

1072 is holomorphic, once we plug in $\mathcal{H}$ from (D.1) in the expression of the conserved current
1073 components (4.2). So the current used in the literature is just a linear combination of
1074 currents defined in our current formalism. The ambiguity of combining currents was
1075 discussed in Section 3.3.

1076     Finally, we show that the operators $J\bar{T}$ and $\hat{J}\bar{T}$ are the same, hence the deformed
1077 theories are equivalent. Writing out the definition (4.19), we get

$$
\begin{aligned}
\hat{J}\bar{T} = 2\pi i\hat{J}_{[t|}T_{\bar{z}|x]} &= J\bar{T} - 4\pi^3\ell\, T_{\bar{z}[t}T_{\bar{z}|x]} \\
&= J\bar{T}\,,
\end{aligned}
\tag{D.3}
$$

1078 which is just the manifestation of the simple fact that the bilinear composite operator built
1079 from the same current is identically zero, $\mathcal{O} \equiv \epsilon^{\mu\nu}J_\mu J_\nu = 0$.

## 1080 E   Quantum perturbation theory formulas

1081 We collect here some formulas relevant to Section 5.2. Local operators are expanded in
1082 modes as

$$
\mathcal{O}(x) = \left(\frac{2\pi}{L}\right)^\Delta \sum_{n=-\infty}^{\infty} e^{2\pi inx/L}\mathcal{O}_n
\tag{E.1}
$$

1083 where $\mathcal{O}_n$ are dimensionless and $\Delta$ is the dimension of $\mathcal{O}(x)$. We denote the CFT (anti-
1084 )holomorphic current and stress tensor by $J, T, \bar{T}$ (signs and factors of $i$ chosen to match
1085 standard 2d CFT literature):

$$
\begin{aligned}
T &= -\left(\frac{2\pi}{L}\right)^2 \sum_{n=-\infty}^{\infty} e^{2\pi inx/L}\,\ell_n, \\
\bar{T} &= -\left(\frac{2\pi}{L}\right)^2 \sum_{n=-\infty}^{\infty} e^{2\pi inx/L}\,\bar{\ell}_n, \\
J &= -i\frac{2\pi}{L} \sum_{n=-\infty}^{\infty} e^{2\pi inx/L}\,j_n,
\end{aligned}
\tag{E.2}
$$

and we recall $T_{tt}^{\mathrm{CFT}} = -T_{xx}^{\mathrm{CFT}} = (T + \bar{T})/(2\pi)$ and $T_{xt}^{\mathrm{CFT}} = T_{tx}^{\mathrm{CFT}} = (T - \bar{T})/(2\pi i)$ and $J_t^{\mathrm{CFT}} = i J_x^{\mathrm{CFT}} = iJ$.

We shifted $\ell_m \equiv L_m - \delta_{m,0}\,c/24$ and $\bar{\ell}_m \equiv \overline{L}_m - \delta_{m,0}\,c/24$ compared to the usual Virasoro algebra, and these modes have non-zero commutators

$$[\ell_m, \ell_n] = (m - n)\ell_{m+n} + \frac{c}{12}m^3\delta_{m+n,0}, \quad [\ell_m, j_n] = -nj_{m+n},$$
$$[\bar{\ell}_m, \bar{\ell}_n] = (m - n)\bar{\ell}_{m+n} + \frac{c}{12}m^3\delta_{m+n,0}, \quad [j_m, j_n] = m\delta_{m+n}. \tag{E.3}$$

After deformation by $J\bar{T}$ we find spectrum-generating operators (we do not display order $\lambda^2$ terms in $\Lambda_k$ and $\overline{\Lambda}_k$ because they are too long)

$$\Upsilon_k = j_k + \delta_{k\neq 0}\frac{2\pi^2 i\lambda}{(1+b)L}\bar{\ell}_{-k} + \delta_{k\neq 0}\left(\frac{2\pi^2 i\lambda}{(1+b)L}\right)^2\left((j_0 + La)\bar{\ell}_{-k} + \frac{1}{2}\sum_{m\neq 0}\frac{m-k}{m}j_m\bar{\ell}_{m-k}\right) + O(\lambda^3),$$
$$\Lambda_k = \ell_k + \frac{2\pi^2 i\lambda}{(1+b)L}\sum_{m\neq 0}j_{k+m}\bar{\ell}_m + O(\lambda^2),$$
$$\overline{\Lambda}_k = \bar{\ell}_k + \frac{2\pi^2 i\lambda}{(1+b)L}\left(\frac{c}{12}k^2 j_{-k} + \sum_{m\neq 0}\frac{m-k}{m}j_m\bar{\ell}_{k+m}\right) + O(\lambda^2). \tag{E.4}$$

Note that $\Upsilon_0 = j_0$ and $\Lambda_0 - \overline{\Lambda}_0 = \ell_0 - \bar{\ell}_0$ as expected. Calculating commutators confirms that these operators obey the same algebra (5.10) as the original modes.

For the $J\bar{T}$ deformation we find

$$H = \frac{2\pi}{L}\left(\frac{\Lambda_0}{1-b} + \frac{\overline{\Lambda}_0}{1+b} + \frac{aL\Upsilon_0}{1-b} + \frac{a^2L^2}{2(1-b)}\right) + 2\pi i\lambda\left(\frac{2\pi}{L}\right)^2\frac{(\Upsilon_0 + aL)\overline{\Lambda}_0}{(1-b)(1+b)^2}$$
$$+ \frac{(2\pi i\lambda)^2}{2}\left(\frac{2\pi}{L}\right)^3\frac{(\Upsilon_0 + aL)^2\overline{\Lambda}_0 + \overline{\Lambda}_0^2/2}{(1-b)(1+b)^3} + O(\lambda^3) \tag{E.5}$$

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
