# Peer review of "Solving a family of $T\bar{T}$-like theories"

_SciPost Physics_

## Round 1 · Referee Report · Anonymous (Referee 1) · 2019-6-16

Strengths

1- the article is well written
2- the subject is very interesting
3- the strategy implemented, and carefully described in section 2, is original

Weaknesses

1- section 3.4 is inconclusive
2- the method used is powerful but no explicit use of quantum integrability was made.

Report

This paper deals with irrelevant integrable deformations of conformal field theories. It aims to generalize recent results concerning the perturbation of 2D models with the Zamolodichikov's TTbar operator and, in particular, to find an exact evolution equation for the spectrum of CFTs perturbed by combinations of specific conserved currents. The authors' strategy, carefully described in section 2, is original and relatively robust.

The main result of the paper is the classical evolution equation encoded in Table 3. The authors made checks to support its validity also at the quantum level. However, there must be a more direct way to arrive at the same flow equation, for example, by working directly within the quantum exact S-matrix setup. Maybe the authors should mention this possibility in the "Conclusions and outlook" section.

In conclusion, this is a serious piece of work in a fast-developing research field, and I recommend this article for publication.

---

## Round 1 · Referee Report · Anonymous (Referee 2) · 2019-6-17

Strengths

1- the subject is interesting

2- novel strategy for solving for the spectrum of $T\bar T$ and $J\bar T$ deformed CFTs, which generalizes the previous approaches

3-the authors have put significant effort in explaining their method

Weaknesses

1- the presentation is oftentimes cumbersome and physically non-transparent

2- the physical significance of the calculations in section 5.2 is not clear

3 - section 3.4 is inconclusive

Report

This article studies irrelevant deformations of 2d QFTs by composite operators constructed from conserved currents, which have the remarkable property that the finite-size spectrum of the deformed theory is exactly solvable. The current article introduces a new method, based on coupling to background fields, which allows to solve for the spectrum in presence of several simultaneous deformations, thus generalizing the previous results in the literature. This method is in principle also applicable to non-conformal theories.

While most of the paper appears physically sound, it is not at all clear what is the physical motivation for introducing the "spectrum-generating operators'' of section 5.2. According to their definition on line 578, these operators are simply the original CFT Virasoro/Kac-Moody generators conjugated by the deformation. Their properties, including the commutation relations (5.10), follow trivially from their definition, and have nothing to do with the symmetries of the deformed theory. The calculation of deformed OPEs using these operators is extremely cumbersome, and it is not clear what are the advantages, if any, of proceeding this way.

In conclusion, this paper introduces a new method to solve a problem of current interest and the main result looks physically robust. At times, the presentation appears unnecessarily complicated and could be improved. I am happy to recommend this article for publication in SciPost, provided the changes below are considered.

Requested changes

1- line 3: since the $T\bar T$ deformation was simultaneously introduced by two different groups, the authors should cite both or neither
2- line 51: it is not true that $T\bar T$-type deformations are not plagued by the usual ambiguities, see e.g. discussion in 1205.6805 and 1603.00719
3- line 121: this statement is only true in absence of anomalies
4- eqn (3.3): since the authors are introducing a different method than previously used in the literature (studying the currents, rather than the conserved charges), it would be helpful if they included the details of the derivation here
5- since (3.2) simply amounts to coupling the CFT to a background vielbein, is there a simple way to understand the deformed solution (3.3) from this point of view, including the trace equation on line 180?
6-line 282: can the authors spell out here what $O_{Ii}$ stands for?
7- the presentation in section 4.1 seems unnecessarily complicated, as the differential operators $D_1$ and $D_2$ are simply implementing the definitions (4.3) and (4.4). Can the authors perhaps simplify the discussion accordingly?
8 - line 382: the results of this table appear to follow from the current commutation relations
9- line 448: this statement is not true in presence of anomalies
10 - section 5.2: can the authors explain what is the physical significance of the spectrum-generating operators and why it was necessary to introduce them?
11 - the statement in footnote 5 is incorrect, as the generators in those papers act in the deformed theory; in particular, their zero modes compute the deformed conserved charges, unlike the zero modes $\Lambda_0, \Upsilon_0$

---

## Editorial Decision

awaiting_resubmission